# Persistent flat band splitting and strong selective band renormalization in a kagome magnet thin film

Zheng Ren [1,11], Jianwei Huang[1,11], Hengxin Tan [2], Ananya Biswas[1], Aki Pulkkinen [3], Yichen Zhang [1], Yaofeng Xie [1], Ziqin Yue[1,4], Lei Chen [1], Fang Xie[1], Kevin Allen [1], Han Wu [1], Qirui Ren[1], Anil Rajapitamahuni[5], Asish K. Kundu[5], Elio Vescovo[5], Junichiro Kono [1,6,7,8], Emilia Morosan[1,8,9], Pengcheng Dai [1,8], Jian-Xin Zhu [10], Qimiao Si [1,8], Ján Minár [3], Binghai Yan [2] & Ming Yi [1,8] ✉

Magnetic kagome materials provide a fascinating playground for exploring the interplay of magnetism, correlation and topology. Many magnetic kagome systems have been reported including the binary $Fe_mX_n$ (X = Sn, Ge; $m{:}n$ = 3:1, 3:2, 1:1) family and the rare earth $RMn_6Sn_6$ (R = rare earth) family, where their kagome flat bands are calculated to be near the Fermi level in the para-magnetic phase. While partially filling a kagome flat band is predicted to give rise to a Stoner-type ferromagnetism, experimental visualization of the magnetic splitting across the ordering temperature has not been reported for any of these systems due to the high ordering temperatures, hence leaving the nature of magnetism in kagome magnets an open question. Here, we probe the electronic structure with angle-resolved photoemission spectroscopy in a kagome magnet thin film FeSn synthesized using molecular beam epitaxy. We identify the exchange-split kagome flat bands, whose splitting persists above the magnetic ordering temperature, indicative of a local moment picture. Such local moments in the presence of the topological flat band are consistent with the compact molecular orbitals predicted in theory. We further observe a large spin-orbital selective band renormalization in the Fe $d_{xy} + d_{x^2-y^2}$ spin majority channel reminiscent of the orbital selective correlation effects in the iron-based superconductors. Our discovery of the coexistence of local moments with topological flat bands in a kagome system echoes similar findings in magic-angle twisted bilayer graphene, and provides a basis for theoretical effort towards modeling correlation effects in magnetic flat band systems.

Quantum solids consisting of the kagome lattice have recently stimulated a surge of interest owing to the rich landscape of quantum phases, potentially driven by the inherent Dirac band crossings, Van Hove singularities (VHSs) and topological flat bands[1–5]. Non-magnetic transition-metal-based kagome metals such as $AV_3Sb_5$ (A = Cs, K, Rb)[6–10], $CsTi_3Bi_5$[11–13] and $ScV_6Sn_6$[14–18] have been shown to host an ensemble of quantum states, including superconductivity, charge density waves (CDW) and nematicity. Complementary to these non-magnetic systems, magnetic kagome materials, such as $Fe_mX_n$ (X = Sn, Ge; $m{:}n$ = 3:1, 3:2, 1:1) and $RMn_6Sn_6$ (R = rare earth), have distinct

potential for realizing other topological and symmetry-breaking phases[19–28]. For example, the gapped Dirac cones lead to a realization of the Chern insulator phase[19,20]. The Weyl semimetal phase and tunable Weyl points have been found in select magnetic kagome materials[21,22]. Interestingly, a novel CDW phase has recently been found to emerge within the antiferromagnetic (AFM) state in a kagome magnet FeGe[23,24].

A fundamental question about a kagome magnet is the origin of its magnetism. Generically, magnetic ordering in solids can be understood from two contrasting limits. In itinerant magnets, a large density of states at the Fermi level triggers a spin-splitting of the electronic bands via the Stoner mechanism, giving rise to an imbalance of spin up and spin down states. In this scenario, the spin splitting of the bands is expected to disappear across the ordering temperature[29]. In the strong-coupling limit, as often the case for magnetic insulators, the electrons are localized and hence produce local moments. Heisenberg exchange coupling between the local moments leads to the long-range ordering of these moments. In this case, as the local moments survive to well above the ordering temperature, the exchange splitting would not show strong temperature dependence across the ordering temperature, but exhibit a diminishing spin polarization[30–32].

For a kagome lattice where quantum destructive interference produces a flat band, a Hubbard model defined with a half-filled flat band predicts a ferromagnetic ground state, in accordance with the Stoner-type itinerant magnetism[33,34]. Thus, magnetic splitting of the electronic bands across the ordering temperature would be expected (Fig. 1h). On the other hand, persistent splitting across the ordering temperature on a kagome lattice would suggest a local moment scenario (Fig. 1i). However, due to the presence of the topological flat band, the local moment should originate from the non-trivial compact molecular orbitals (Fig. 1j)[35]: This happens when the strength of the Coulomb repulsion ($U$) lies in between the width of the flat band ($D_{flat}$) and that of the wide bands ($D_{wide}$)[36]; as further discussed in Supplementary note 1, it is to be contrasted with the formation of atomic local moments when the interaction $U$ exceeds the width of all the bands. Despite the importance of this question, up to now, there has not been any direct experimental study of the band evolution across the magnetic ordering temperature in any kagome magnet, leaving the nature of magnetism in kagome lattice materials an open question.

Here, we explore this question via the magnetic kagome system FeSn, enabled by our combined molecular beam epitaxy (MBE) and angle-resolved photoemission spectroscopy (ARPES) system that allows us to synthesize robust high-quality FeSn thin films and perform in-situ ARPES measurements. By comparing the experimental data with density functional theory (DFT) calculations, we identify the Dirac crossings and flat bands, in the presence of magnetic splitting in the A-type AFM phase. By varying the temperature from 20 K to above the Néel temperature ($T_N = 370$ K), we observe evident temperature evolution of the band structure qualitatively consistent with the exchange splitting between spin majority and spin minority bands, but with a significant persistent splitting above $T_N$. This suggests the nature of the magnetism to be dominated by the presence of local moments rather than Stoner instability. Interestingly, while most of our data show reasonable match with DFT, we discover a strong band renormalization in a subset of the band structure. We further uncover that the renormalized bands only inhabit the $d_{xy} + d_{x^2-y^2}$ spin majority channel,

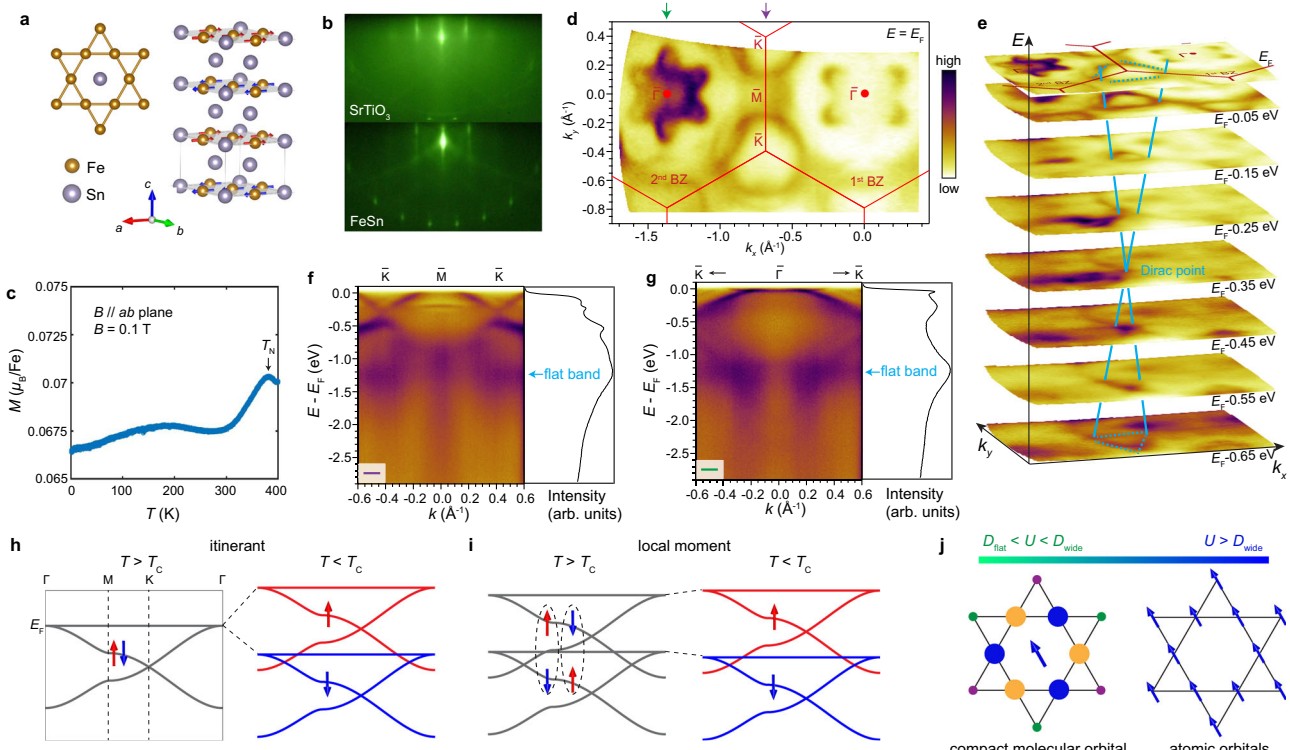

**Fig. 1 | Basic characterizations of the FeSn film. a** Schematic of the crystal and magnetic structure of FeSn. Brown and blue spheres denote the Fe and Sn atoms, respectively. **b** RHEED image of SrTiO₃(111) and the FeSn film. **c** Magnetization as a function of temperature taken with a magnetic field of 0.1 T applied parallel to the *ab*-plane. **d** FS map taken at 45 K overlaid with the BZ boundaries. Green and purple arrows indicate the cuts shown in (**f**, **g**). **e** Constant energy contours for the same *k*-space region as in (**d**). Blue solid and dashed lines denote the Dirac dispersions. **f**, **g** K̄ – M̄ – K̄ and K̄ – Γ̄ – K̄ cuts taken at 83 K and their momentum-integrated EDCs. **h** Schematic of the kagome band splitting across $T_C$ driven by the itinerant flat band magnetism. Red and blue arrows denote the opposite spins. **i** Schematic of persistent band splitting and diminishing spin polarization above $T_C$ in the local moment scenario. Dashed lines indicate two degenerate cases of exchange splitting for spin up and spin down local moments. **j** Schematics of the compact molecular orbital and atomic orbitals and their local moments in different regimes of $U$. Size and color of the filled circles indicate the amplitude and phase of the Wannier function[35].

suggesting a strong spin and orbital selective correlation effect in a magnetic kagome system.

## Results

We start with characterizing the basic properties of our epitaxial FeSn films. FeSn is constituted by alternating $Fe_3Sn$ kagome layers and Sn honeycomb layers, in the space group P6/mmm (Fig. 1a). Previous studies have shown that FeSn is an A-type antiferromagnet, with each Fe kagome layer being ferromagnetic and anti-aligned in the stacking direction. The Fe moments lie in the *ab*-plane (Fig. 1a)[37–39]. We synthesize 30 nm thick FeSn films on the $SrTiO_3$(111) substrates (Methods) and confirm the crystallinity in the reflection high-energy electron diffraction (RHEED) pattern that exhibits sharp zeroth and higher order streaks and Kikuchi lines (Fig. 1b). The spacing between the streaks in FeSn is slightly larger than that in $SrTiO_3$(111), consistent with their lattice mismatch of 4%[40,41]. X-ray diffraction (XRD) further confirms the single phase of FeSn (Supplementary Fig. 2). We further measure the magnetization of a thicker FeSn film which yields the $T_N$ of ~370 K (Fig. 1c), consistent with previous studies on FeSn bulk crystals[42].

After confirming the quality of the FeSn film, we perform in-situ ARPES measurements with a 21.2 eV helium lamp photon source. Previous synchrotron-based ARPES studies on FeSn bulk crystals have identified the Dirac cones and signatures of flat bands[43–46]. In this study, the lower photon energy provides a distinct advantage of higher energy resolution. We also note that the surface quality of our epitaxial thin films is significantly more robust than the cleaved surface of bulk crystals, which enables extended ARPES mapping over the course of a week and multiple thermal cycles between 20 K and 400 K.

Fermi surface (FS) mapping of the FeSn film covering the first and second Brillouin zones (BZs) displays a flower-like contour with six petals around the Γ point (Fig. 1d). We note that part of this feature in the 1st BZ is suppressed due to matrix element effects. A triangular pocket is observed at the K point, which shows a linear dispersion as evident in the stack of constant energy contours down to 0.65 eV below the Fermi level ($E_F$) (Fig. 1d, e). Based on comparison with DFT calculations (Fig. 2c), we identify it as the Dirac cone at the K point, with the Dirac point located at -0.35 eV (Fig. 1e). Furthermore, we identify the largely non-dispersive band that produces a peak in the accumulated density of states (DOS) at -1.2 eV (Fig. 1f, g). This feature matches the location of the kagome flat bands in agreement with the DFT calculations for the AFM state (Fig. 2a, c).

Next, we explore the temperature evolution of the characteristic band features. The A-type AFM structure of FeSn can be viewed as a stack of ferromagnetic (FM) kagome layers with alternating spin directions[24]. An exchange splitting between spin majority and spin

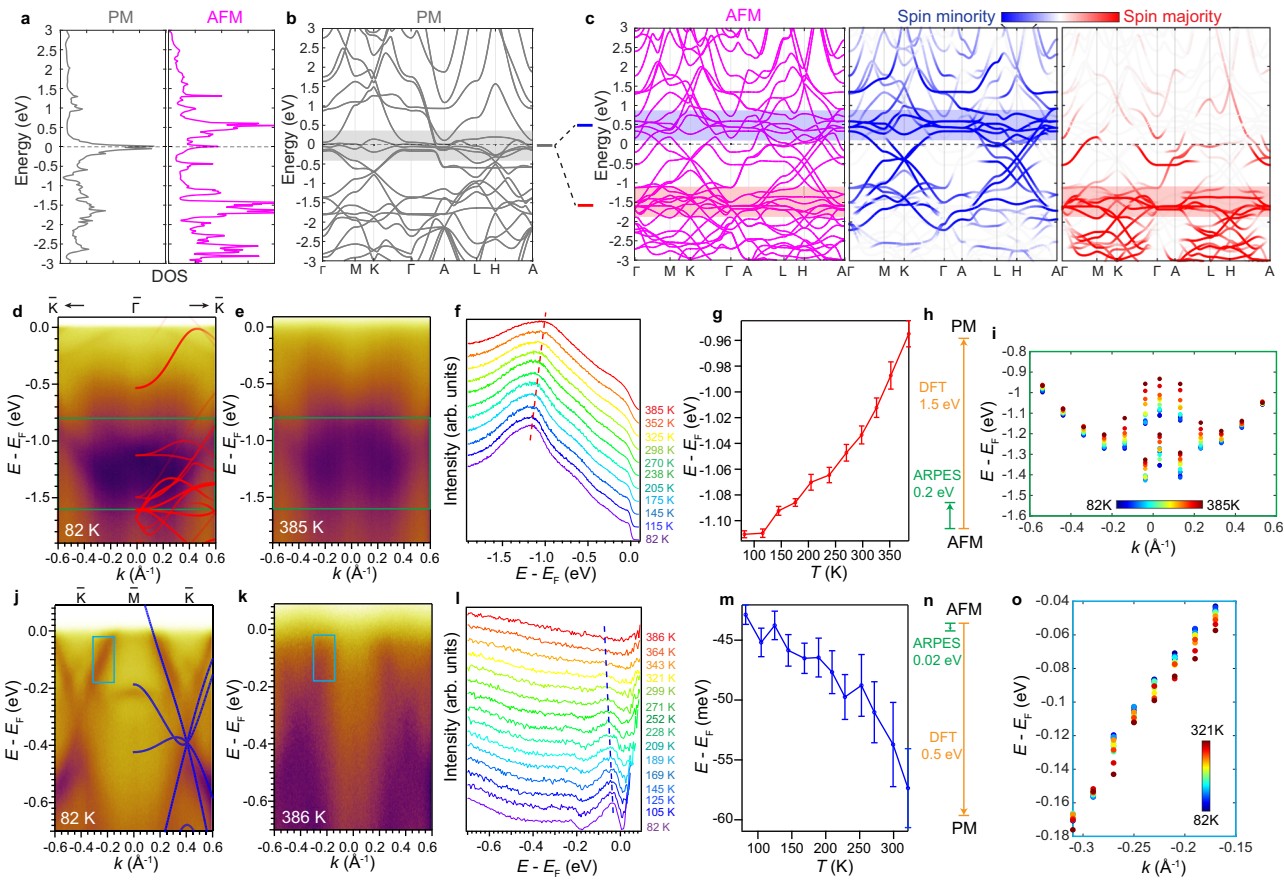

**Fig. 2 | Persistent exchange splitting above $T_N$. a** Calculated DOS distribution for the PM and AFM phases. **b-c** DFT calculations in the PM phase (grey) and the AFM phase (magenta), with the spin majority (red) and spin minority (blue) bands projected for each kagome layer. Grey, blue and red shaded areas denote the flat band regions in the PM, spin minority and spin majority bands, respectively. **d, e** $\bar{K} - \bar{\Gamma} - \bar{K}$ cuts taken at 82 K and 385 K. Spin majority-projected DFT calculations are superimposed on the cut in (**d**). **f** EDCs as a function of temperature taken from the $\bar{K} - \bar{\Gamma} - \bar{K}$ cut at $\bar{\Gamma}$. Red dashed line is a guide to the eye for the shift of the two peaks in the fitting (Supplementary Fig. 3). **g** Energy of the fitted peak closer to $E_F$ in (**f**) as a function of temperature (see details of the fitting in Supplementary

Fig. 3). Error bars are from the standard deviation resulting from the fitting process. **h** Schematic that shows the ratio of observed peak shift in (**g**) and the calculated band shift from PM to AFM phase in DFT. **i** Observation of the momentum-dependent band shifts as a function of temperature in the region enclosed in the green box in (**d**). At each selected momentum the temperature-dependent EDCs are fitted as shown in Supplementary Fig. 3, and the peaks are plotted here as the colored dots. **j–o** Data and analysis for the $\bar{K} - \bar{M} - \bar{K}$ cut corresponding to each panel in (**d–i**). Spin minority-projected DFT bands are superimposed in (**j**). **l, m** are extracted from the cuts at $k = -0.17$ Å$^{-1}$. See Supplementary Fig. 4 for the details of fitting.

minority bands is expected to occur within each layer, as shown in the DFT calculations for the paramagnetic (PM) and AFM phases (Fig. 2a–c). Importantly, the large DOS peak in the PM phase at $E_F$, corresponding to the kagome flat band, splits into two peaks, corresponding to the spin majority and minority flat bands located at $E_F$-1.5 eV and $E_F$ + 0.5 eV, respectively. This understanding is consistent with the A-type AFM structure of FeGe[24]. To evaluate the exchange splitting experimentally, we explore the temperature dependence of electronic dispersions measured along two high-symmetry cuts. The near-$\bar{\Gamma}$ region along $\bar{K} - \bar{\Gamma} - \bar{K}$ and the $\bar{K} - \bar{M} - \bar{K}$ cuts are dominated by spin majority and spin minority bands in the energy range within 1.5 eV below $E_F$, respectively, as shown in the DFT calculations (Fig. 2c). We find the experimental data taken deep in the AFM phase to be mostly consistent with the calculations (Fig. 2d, j). The $\bar{K} - \bar{\Gamma} - \bar{K}$ cut mainly shows the kagome quadratic band bottom at the zone center, and the $\bar{K} - \bar{M} - \bar{K}$ cut shows the two Dirac crossings (Fig. 2d, j). As we increase the temperature from 82 K to 385 K, the bands along $\bar{K} - \bar{\Gamma} - \bar{K}$ shift up in energy, as demonstrated in the stack of energy distribution curves (EDCs) taken near the $\bar{\Gamma}$ point (Fig. 2d–g, Supplementary Fig. 3). This upward shift from the AFM phase to the PM phase is consistent with the spin majority nature of the band assignment. Meanwhile, the Dirac bands in the $\bar{K} - \bar{M} - \bar{K}$ cut exhibits a downward shift as temperature is increased, as shown directly in the temperature-dependent EDCs (Fig. 2j–m, Supplementary Fig. 4), and confirmed by analysis of the temperature-dependent momentum distribution curves (MDCs) (Supplementary Fig. 7). We note that for a more accurate determination of the band positions near $E_F$, the EDCs have been divided by the Fermi-Dirac function at corresponding temperatures before the fitting. The observed band shifts are robust to thermal cycle tests, hence are intrinsic and not due to surface degradation (Supplementary Fig. 5). In addition, we can also eliminate the possibility that these shifts are due to a simple lattice thermal expansion (Supplementary Note 2 and Supplementary Fig. 6), and confirm that lattice expansion is not the major contribution to the observed band shifts.

Although the temperature evolution is evident and clearly demonstrates the spin majority and spin minority nature of the bands, interestingly, the magnitude of the shift is significantly smaller than that expected from the exchange splitting across the PM-AFM phase transition. Specifically, the upward shift of the $\bar{K} - \bar{\Gamma} - \bar{K}$ band bottom is ~0.2 eV and the downward shift of the $\bar{K} - \bar{M} - \bar{K}$ bands near $E_F$ is ~0.02 eV, which are about 14% and 4% of the full exchange energy scale calculated from DFT (Fig. 2h, n), respectively. Furthermore, we evaluate the momentum-dependent EDCs and find that the upward (downward) shift is the largest at the band bottom ($E_F$), while away from these points, at intermediate energies, the shift appears to be smaller (Fig. 2i, o). Therefore, instead of a full merging of the spin majority and minority bands, there is significant portion of persistent splitting above $T_N$, suggestive of the presence of local moments. Such a persistent splitting is also verified for the spin majority flat band (Supplementary Fig. 8).

Having identified the persistent exchange splitting above $T_N$, we next turn to another intriguing aspect of our finding. Although most bands in the DFT calculations have a good match in the ARPES data (Fig. 2d, j), the calculated spin majority electron-like band along $\bar{K} - \bar{\Gamma} - \bar{K}$ within 0.5 eV below $E_F$ seems to have no direct experimental counterpart (Fig. 2d). To examine the origin of these bands, we provide a detailed comparison of our measured dispersions along high symmetry directions with the orbital-projected DFT bands in the AFM phase (Fig. 3). As our measurements are taken with the helium lamp with a single photon energy of 21.2 eV, our location along $k_z$ is in between 0 and $\pi$ (See the discussion on photon-energy dependence measurement in Supplementary Note 3 and Supplementary Fig. 9). We therefore compare our data with both of the extrema $k_z$ planes of 0 ($\Gamma - K - M$) and $\pi$ ($A - L - H$). As the electronic structure near $E_F$ is dominated by Fe 3 d orbitals, we find that our data can be largely

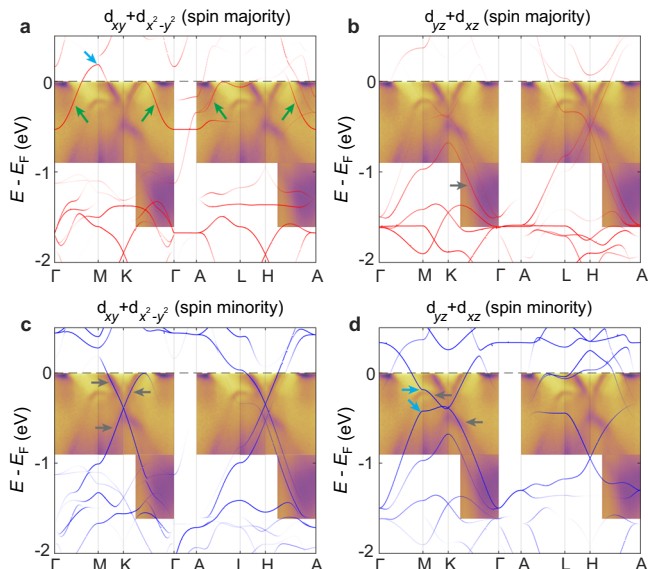

**Fig. 3 | Comparison of ARPES data and DFT calculations in different spin and orbital channels. a–d** $d_{xy} + d_{x^2-y^2}$ and $d_{xz} + d_{yz}$ orbital-projected DFT calculations for spin majority and spin minority bands. $\bar{K} - \bar{\Gamma} - \bar{K}$ and $\bar{K} - \bar{M} - \bar{K}$ cuts are overlaid on the spin majority and spin minority bands, respectively. Green arrows in (**a**) indicate the calculated electron bands that do not match the $\bar{K} - \bar{\Gamma} - \bar{K}$ cut. Grey arrows in (**b–d**) mark the matching parts of the experimental and calculated bands. Blue arrows in (**a–d**) mark the calculated VHSs.

captured by the Fe $d_{xy} + d_{x^2-y^2}$ and $d_{xz} + d_{yz}$ orbitals. In particular, the measured dispersions match the bands from the $d_{xy} + d_{x^2-y^2}$ spin minority and $d_{xz} + d_{yz}$ spin majority/minority bands (Fig. 3b-d). However, two aspects show large deviations from the calculations. First, kagome lattice should produce VHSs at the M point of the BZ, as has been observed in AFM FeGe and AV$_3$Sb$_5$ (A = Cs, K, Rb). In the AFM calculation for FeSn near $E_F$ (blue arrows in Fig. 3), we see that the DFT calculation shows a VHS at the M point above $E_F$ that is destroyed due to hybridization of the $d_{xy} + d_{x^2-y^2}$ spin majority and minority bands (Fig. 3a). Below $E_F$, another pair is located at -0.2 eV and -0.4 eV at the M point, dominantly of $d_{xz} + d_{yz}$ orbital. It is important to note that as $d_{xz}$ and $d_{yz}$ are three-dimensional orbitals and these VHS exhibit strong $k_z$-dispersion such that along A-L they no longer preserve their saddle-point behavior. In the measured dispersions, a band is indeed observed near the location of the higher VHS, but it is hole-like along both $\bar{\Gamma} - \bar{M}$ and $\bar{M} - \bar{K}$, and is therefore not a VHS, likely due to this strong $k_z$-dispersion. The second aspect that strongly deviates from the calculations are the extremely narrow bands observed near $\bar{\Gamma}$, as marked by the green arrows in Fig. 3a.

To investigate the origin of these narrow bands, we examine the dispersions in the second BZ associated with the aforementioned flower-like FS at $\bar{\Gamma}$ (Fig. 1d). Remarkably, there are two electron-like bands with an extremely small bandwidth, as shown on the $\bar{K} - \bar{\Gamma} - \bar{K}$ cut in the 2nd BZ (Fig. 4a), one forming the flower-like pocket while the other a circular pocket. With a 0.8 Å$^{-1}$ momentum span, this feature is restricted within a 40 meV energy window below $E_F$, leading to a large effective mass $m^* \approx 7m_e$, as extracted from a parabolic fitting of the dispersion. The electron-like bands have a similar Fermi momentum ($k_F$) with the electron-like bands in DFT (green arrows in Fig. 3a), albeit with a much narrower bandwidth. This is suggestive of a strong renormalization effect selectively occurring to these electron-like bands.

To gain further insight into this conjecture and validate that the band renormalization is beyond the consideration of DFT, we first cross check predictions between different methods of DFT

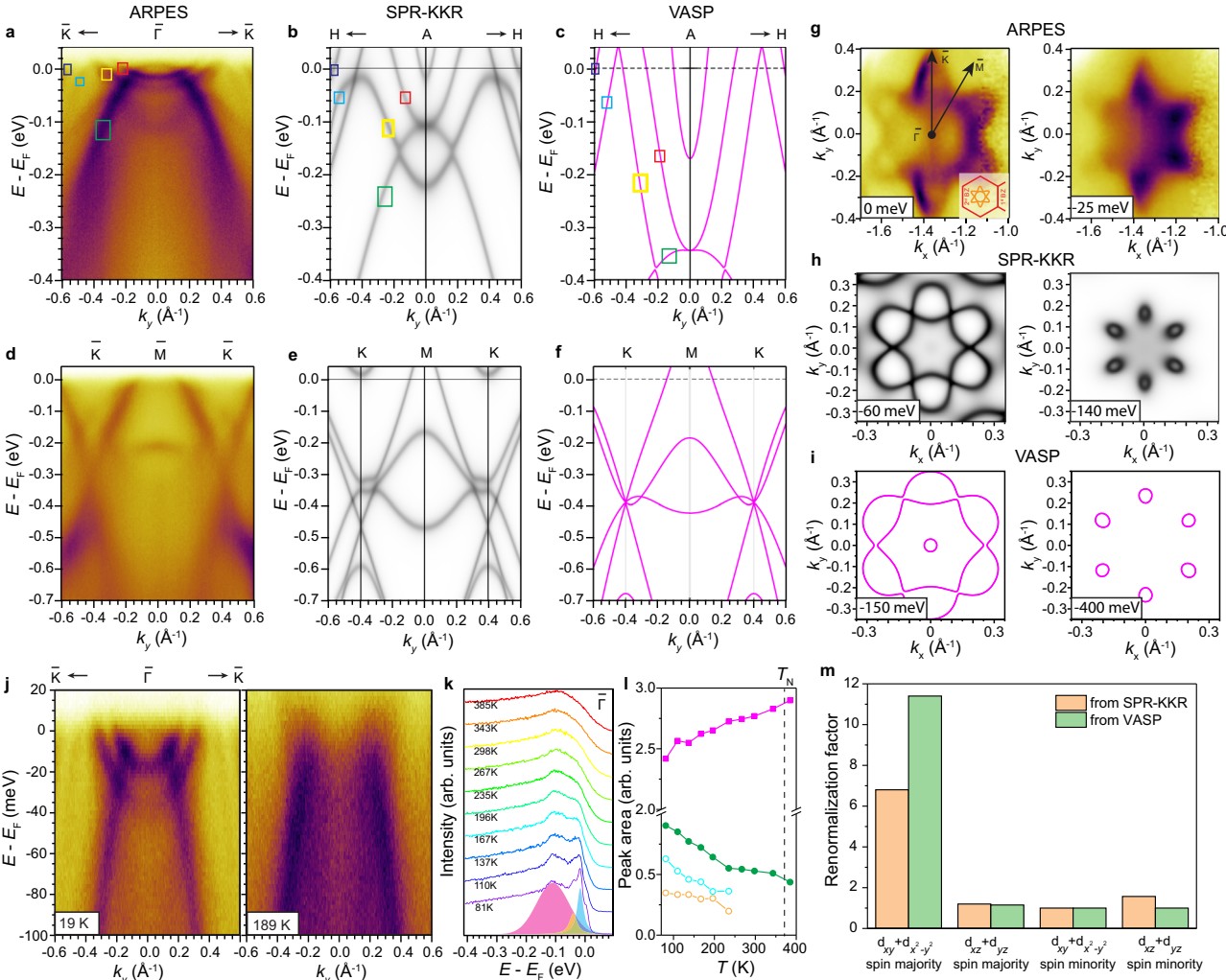

**Fig. 4 | Strong band renormalization in $d_{xy} + d_{x^2-y^2}$ spin majority channel.**
**a**–**c** Comparison of the $\bar{K} - \bar{\Gamma} - \bar{K}$ cut in ARPES data, SPR-KKR calculations and VASP calculations ($k_z = \pi$, see Supplementary Note 3). Colored boxes help identify the corresponding bands in ARPES data and calculations. **d**–**f** Comparison of the $\bar{K} - \bar{M} - \bar{K}$ cut in ARPES data, SPR-KKR calculations and VASP calculations. **g** CECs extracted from ARPES data at binding energies of 0 meV and 25 meV, showing the change of the flower-shaped electron pockets. **h**, **i** Calculated CECs using SPR-KKR and VASP, respectively, showing similar features as in **g**. The binding energies are as shown. **j** $\bar{K} - \bar{\Gamma} - \bar{K}$ cuts in a smaller energy window taken at 19 K and 189 K. **k** EDCs as a function of temperature taken at $k_y = 0$ in (**j**) Magenta, yellow and blue shaded regions are the fits of the peaks. **l** Peak area (proportional to spectral weight) of the fitted peaks of the same color in **k**. Above 235 K blue and yellow peaks can no longer be distinguished, so green corresponds to the sum of blue and yellow peaks. **m** Band renormalization factors in different spin-orbital channels, from SPR-KKR (yellow) and VASP (green) band calculations.

implementation, namely the pseudopotential approach of Vienna ab-initio Simulation Package (VASP) and the all-electron full-potential fully-relativistic treatment in the spin polarized relativistic Korringa-Kohn-Rostoker (SPR-KKR) package. Intriguingly, the electron-like bands close to $E_F$ at Γ show the largest renormalization effect compared to both DFT methods (SPR-KKR and VASP). In addition, we also carried out dynamic mean-field theory (DMFT) calculations, where the renormalization trend is consistent (green boxes in Supplementary Fig. 10). A side-by-side comparison between ARPES data, SPR-KKR and VASP calculations illustrates the consistency in the Dirac cones (Fig. 4d–f). However, the two electron-like bands at Γ become progressively renormalized from VASP to SPR-KKR to ARPES, while the hole-like band only shifts towards $E_F$ but is not renormalized much (Fig. 4a–c). Remarkably, a comparison of the constant energy contours (CECs) between ARPES data, SPR-KKR and VASP calculations taken at correspondingly renormalized binding energies shows a striking resemblance, where the flower petals shrink but maintain a hexagonal layout as the binding energy increases (Fig. 4g–i). These observations provide strong evidence for a picture of spin and orbital selective

renormalization effect: $d_{xy} + d_{x^2-y^2}$ spin majority channel is strongly renormalized (flower-shaped electron-like bands at Γ); $d_{xz} + d_{yz}$ spin minority channel (hole-like band at Γ (Fig. 3d)) and $d_{xy} + d_{x^2-y^2}$ spin minority channel (Dirac cones at K) show almost negligible renormalization effect. Remarkably, the renormalization factor in the $d_{xy} + d_{x^2-y^2}$ spin majority channel is as large as 6.8 from SPR-KKR and 11.4 from VASP (Fig. 4m).

Interestingly, the spectral weight of the strongly renormalized bands rapidly diminishes as the temperature is raised, as seen in the comparison of $\bar{K} - \bar{\Gamma} - \bar{K}$ cut taken at 19 K and 189 K (Fig. 4j). We fit the EDC at the $\bar{\Gamma}$ point taken at different temperatures and extract the spectral weight of the two shallow bands, which drastically decreases as the temperature increases (Fig. 4k, l, green). In contrast, the higher energy band at -0.1 eV does not show such spectral weight suppression (Fig. 4k, l, magenta).

Given the remarkable spin/orbital-selective renormalization of these electron bands, we have also performed three types of rigorous checks to rule out the possibility that they derive from surface states. First, we have performed DFT slab calculations of surface states, which

fail to reproduce the strongly renormalized electron-like bands (Supplementary Note 4 and Supplementary Fig. 11). Second, we have tested the robustness of these bands by capping our MBE-grown films and de-capping at a synchrotron, where we can reproduce the observed shallow electron bands. This test together with thermal cycling tests demonstrate the remarkable robustness of these states, suggesting that they are bulk states[45,47] (Supplementary Note 5 and Supplementary Fig. 12). Third, we tested the universality of these states by measuring cleaved single crystals of FeSn, where the strongly renormalized electron-like bands can be observed on both the Sn termination and the kagome termination, indicative of their bulk nature (Supplementary Note 6 and Supplementary Fig. 14). Here we note that the prominent flower shape of the electron pocket observed on thin films does not seem to be observed on FeSn single crystals. Although our data provide strong evidence that the remarkably renormalized bands are bulk states, we note the apparent conflicting evidence for them being surface states in a previous study on FeSn single crystals[43]. To resolve this, future work including photon energy dependent measurements as well as studies to explore the effect of dimensionality would be desired. Nevertheless, the robustness of the strong renormalization of these states are observed on all electron bands measured on thin films as well as single crystals.

## Discussion

Our experiments reveal a series of intriguing phenomena related to band topology and electronic correlation in the kagome magnet FeSn. First, we identify the topological flat band resulting from the destructive interference in the AFM phase of FeSn. Our observation of the flat bands is consistent with the expected location of the spin majority kagome flat band from DFT calculations for the AFM phase, suggesting that the DFT estimation of the exchange splitting in the AFM phase is largely reasonable. However, the spin-split flat bands are found to remain split above $T_N$. This is consistent with the behavior of magnetism driven largely by local moments, in contrast to the Stoner-type flat band magnetism[33,34]. Importantly, we note that the topological nature of the kagome flat bands in principle prohibits the representation of the electronic states using localized atomic Wannier orbitals[2,48], contrasting our finding of the coexistence of local moments and topological flat bands. Instead, this may be consistent with the treatment of the compact molecular orbitals that effectively could act like local moments in analogy to the atomic local moments, as has been theoretically advanced for the case of kagome systems[35] that exemplify bulk frustrated lattice materials (Supplementary note 1). This cross-links with the case of magic-angle twisted bilayer graphene. There, seemingly contradictory observations of localized moments and itinerant electrons have been reported, in the presence of a topological flat band at $E_F$[49,50], and theoretical treatment of continuum models has been proposed to reconcile the construction of localized states with topological flat bands[51]. Interestingly, a recent inelastic neutron scattering study finds anomalous high-energy magnetic modes consistent with spin clusters associated with the localized flat band excitations in another kagome magnet TbMn$_6$Sn$_6$[52], suggesting a similar scenario as in our findings.

Second, we discover a strong selective band renormalization in a particular spin and orbital channel. The selective renormalization together with the rapid depletion of the spectral weight of the renormalized bands with increasing temperature is reminiscent of the orbital-selective correlations observed in multiorbital systems, most prominently reported in the ruthenates[53–55] and the iron-based superconductors[56–62]. In these systems deemed Hund's metals, bands associated with a particular orbital is strongly renormalized already at low temperatures, with mass enhancement ranging from 25 in the case of ruthenates and up to 40 in the iron chalcogenides, but retaining relatively well-defined electronic states. Above a characteristic temperature scale, these strongly renormalized orbitals

lose coherence and are no longer well-defined quasiparticles. Theoretically, this behavior has been understood to arise from a combination of Hund's coupling $J$ and Coulomb interaction $U$. For such multiorbital systems away from half-filling, the occupation of different orbitals could vary, with some closer to half-filling. As demonstrated by both slave-boson calculations and dynamical mean field theory calculations, these orbitals are typically more strongly renormalized and exhibit a lower coherence temperature scale than other orbitals, where photoemission measurements would observe a spectral weight depletion for these orbitals as a function of temperature at a much lower temperature than other less renormalized orbitals[63–68]. Here in the case of FeSn, both characteristic strong selective renormalization as well as coherence depletion are observed for the electron bands near $\bar{\Gamma}$. Interestingly, spin also participates in the selectivity as another degree of freedom in addition to orbital. Future theoretical investigation is desired to understand the spin-orbital selectivity in FeSn.

Lastly, our high quality ARPES data indicate the absence of VHSs near $E_F$ in AFM FeSn. Such a scenario contrasts with the VHSs observed near $E_F$ in the isostructural A-type AFM FeGe, where a $2 \times 2$ CDW has been observed[24]. In FeGe where the exchange splitting of the bands with the same ferromagnetic kagome layers pushes the spin majority VHSs to near the $E_F$, the conditions for the theoretical proposal of nesting-mediated CDW via the VHSs at the M points of the BZ are fulfilled. While it is unlikely that VHSs alone are able to drive a CDW in these bulk kagome systems, the lack of VHSs near $E_F$ in FeSn and the lack of CDW order in contrast to the isostructural FeGe may still indicate a necessary but insufficient requirement of VHSs for the presence of $2 \times 2$ CDW order in kagome lattices.

Overall, our results suggest that electron correlations effects are non-negligible in iron-based and likely manganese-based kagome magnets. Recent theoretical efforts have started to consider the effect of correlations in mapping the kagome metals to the Mott insulating limit of quantum spin liquid candidates[69]. Such effects strongly affect the type of complex magnetic and charge symmetry breaking orders in these systems, and may play an important role in the intertwinement of such orders within the same system[23,24]. In this direction, our work on FeSn may play a benchmark for guiding theoretical efforts at gauging the strength of correlations for better understanding of the emergent phases in the class of metallic kagome magnets at large.

## Methods

### Molecular Beam Epitaxy (MBE) growth

Buffered hydrofluoric acid treated niobium-doped (0.05 wt%) SrTiO$_3$(111) substrates (Shinkosha) were cleaned in an ultrasonic bath of acetone and 2-propanol, and then loaded into our custom-built MBE chamber (base pressure $1 \times 10^{-9}$ Torr). The substrates were heated up to the growth temperature at around 600 °C and degassed. Fe and Sn were co-evaporated from separate Knudsen cells stabilized at 1240 °C and 1000 °C, respectively. The flux ratio was calibrated to be ~1:1 using a quartz crystal microbalance. The growth rate was nominally 2.5 min per unit cell thickness. 15 kV reflection high energy electron diffraction was used to monitor the growth. After the growth, the samples were cooled back to room temperature and transferred in-situ to our ARPES chamber.

### ARPES measurements

ARPES measurements (except Supplementary Fig. 9, Supplementary Fig. 12c, d, Supplementary Fig. 13 and Supplementary Fig. 14) were carried out at Rice University equipped with a helium lamp ($h\nu = 21.2$ eV, Fermion Instruments) and a SCIENTA DA30 electron analyzer with a base pressure of $5 \times 10^{-11}$ Torr. Temperatures at which the data were taken are indicated in figures and captions. ARPES measurements shown in Supplementary Fig. 9, Supplementary Fig. 12c, d, Supplementary Fig. 13 and Supplementary Fig. 14 were

carried out at ESM (21ID-I) beamline of the National Synchrotron Light Source II using a SCIENTA DA30 analyzer.

## First-principle calculations

**Vienna ab-initio Simulation Package (VASP)[70].** The electron-electron exchange interaction is mimicked with the generalized gradient approximation (GGA) parametrized by Perdew-Burke-Ernzerhof[71]. The FeSn crystal structure is fully relaxed under the AFM configuration until the maximal remaining force on atoms is no larger than 1 meV/Å. An energy cutoff of 350 eV is used for plane wave basis set. A $k$-mesh of $12 \times 12 \times 6$ is employed to sample the reciprocal space. The spin-orbital coupling effect is negligible and thus not considered throughout. Notice that the orbital and spin resolved band structures in the main text are obtained by projecting the total band structure to one of the two kagome layers of the AFM phase.

## Spin Polarized Relativistic Korringa-Kohn-Rostoker (SPR-KKR)

The Bloch spectral function was calculated using the fully relativistic, full potential Korringa-Kohn-Rostoker method based on multiple scattering and Green's functions, as implemented in the SPRKKR package[72]. Relativistic effects are described by the Dirac equation. The exchange and correlation effects were treated at the level of local spin density approximation (LSDA), and basis set is truncated at $l_{max} = 3$.

## Dynamical Mean Field Theory (DMFT)

A charge self-consistent combination of DFT with DMFT[73,74] calculations were performed with a full-potential linearized augmented plane wave as implemented in the WIEN2k code[75]. The generalized gradient approximation (GGA)[71] was used for the exchange-correlation functional. The spin-orbit coupling was not included in the calculation. The muffin-tin radius $2.49a_0$ ($a_0$ being the Bohr radius), $2.34a_0$ for Fe and Sn respectively, and a plane wave cutoff $RK_{max} = 8$ were taken in calculations that included $15 \times 15 \times 8$ **k**-points. Within DFT + DMFT, we used $U_{Fe} = 5.0$ eV and Hund's rule interactions $J = 0.79$ eV to get insight into the role of electronic correlations on the electronic structure in the present antiferromagnetic system at $T = 58$ K. For the DMFT, a strong-coupling version of continuous-time quantum Monte Carlo (CT-QMC) method[76–78], which provides numerically exact solutions, was used to solve the effective multiple-orbital quantum impurity problem self-consistently.

## Data availability

All data needed to evaluate the conclusions are present in the paper and Supplementary Information. Additional data are available from the corresponding author upon request.

## Code availability

The band structure calculations used in this study are available from the corresponding author upon request.

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

## Acknowledgements

This research used resources of the National Synchrotron Light Source II, a U.S. Department of Energy (DOE) Office of Science User Facility operated for the DOE Office of Science by Brookhaven National Laboratory under Contract No. DE-SC0012704. The single-crystal synthesis work at Rice was supported by the U.S. DOE, BES under Grant No. DE-SC0012311 (P.D.) and the Robert A. Welch foundation grant number C-1839 (P.D.). The ARPES work at Rice University was supported by the U.S. DOE grant No. DE-SC0021421. The MBE work was supported by the Gordon and Betty Moore Foundation's EPiQS Initiative through grant No. GBMF9470 and the Robert A. Welch Foundation Grant No. C-2175 (M.Y.). Z.R. was partially supported by the Rice Academy of Fel-lows program. Y.Z. is partially supported by the Air Force Office of

Scientific Research (AFOSR) Grant No. FA9550-21-1-0343. The bulk characterization measurements were supported by the Department of Defense, Air Force Office of Scientific Research under Grant No. FA9550-21-1-0343, with partial support for E.M. from the Robert A. Welch Foundation Grant No. C-2114. The theory work at Rice is supported primarily by the U.S. Department of Energy, Office of Science, Basic Energy Sciences, under Award No. DE-SC0018197 (L.C.), by the AFOSR Grant No. FA9550-21-1-0356 (F.X.), by the Robert A. Welch Foundation Grant No. C-1411 (Q.S.), and by the Vannevar Bush Faculty Fellowship ONR-VB N00014-23-1-2870 (Q.S.). J.M. and A.P. would like to thank the QM4ST project with Reg. No. CZ.02.01.01/00/22_008/0004572, cofunded by the ERDF as part of the MŠMT. Work at Los Alamos was carried out under the auspices of the U.S. DOE National Nuclear Security Administration (NNSA) under Contract No. 89233218CNA000001 and was supported by LANL LDRD Program, and in part by the Center for Integrated Nanotechnologies, a DOE BES user facility. B.Y. acknowledges the financial support by the Israel Science Foundation (ISF: 2932/21, 2974/23), German Research Foundation (DFG, CRC-183, A02), and by a research grant from the Estate of Gerald Alexander. We acknowledge support from the US National Science Foundation (NSF) Grant Number 2201516 under the Accelnet program of Office of International Science and Engineering (OISE). J.K. acknowledges support from the Robert A. Welch Foundation (C-1509).

## Author contributions

M.Y. supervised the project. Z.R. and A.B. grew the MBE films. Y.X. grew the single crystals under the guidance of P.D.. Z.R., J.H., A.B., Y.Z., Z.Y., H.W., Q.R. performed the ARPES measurements and data analysis under the guidance of M.Y. and J.K.. A.R., A.K., and E.V. assisted and supported the synchrotron ARPES measurements. H.T. and B.Y. performed the VASP calculations. A.P. and J.M. performed the SPR-KKR calculations. J.Z. performed the DMFT calculations. L.C., F.X. and Q.S. provided theoretical model inputs. Z.R. and A.B. performed the XRD and magnetization measurements with the help of K.A. and E.M.. Z.R. and M.Y. wrote the manuscript with the help from all co-authors.

## Competing interests

The authors declare no competing interests.

## Additional information

[1]Department of Physics and Astronomy, Rice University, Houston, TX 77005, USA. [2]Department of Condensed Matter Physics, Weizmann Institute of Science, Rehovot, Israel. [3]New Technologies-Research Center, University of West Bohemia, Plzeň 301 00, Czech Republic. [4]Applied Physics Graduate Program, Smalley-Curl Institute, Rice University, Houston, TX 77005, USA. [5]National Synchrotron Light Source II, Brookhaven National Lab, Upton, NY, USA. [6]Department of Electrical and Computer Engineering, Rice University, Houston, TX 77005, USA. [7]Department of Materials Science and NanoEngineering, Rice University, Houston, TX 77005, USA. [8]Smalley-Curl Institute, Rice University, Houston, TX 77005, USA. [9]Department of Chemistry, Rice University, Houston, TX 77005, USA. [10]Theoretical Division and Center for Integrated Nanotechnologies, Los Alamos National Laboratory, Los Alamos, NM, USA. [11]These authors contributed equally: Zheng Ren, Jianwei Huang. ✉e-mail: mingyi@rice.edu

