## [Peer Review File · Nature Communications]

Editorial Note: Figure R3 on Page 25 in this Peer Review File is reproduced with permission from Nature Communications.

REVIEWER COMMENTS

Reviewer #1 (Remarks to the Author):

FeSn is a magnetic kagome compound, which raises important questions about the coexistence of magnetism, correlations and topology. This manuscript reports an ARPES study of its electronic structure across the magnetic transition at $T_N=370\text{K}$. Although there have already been a number of ARPES studies of this system (41,42, but Moore et al. PRB 106, 115141 (2022) and Multer Communication Materials 4, 17 (2023) should also be quoted), it was never up to the magnetic transition. Understanding how the electronic structure evolves across the transition is an important issue as it should characterize the type of electronic correlations.

The ARPES experiment is performed on in-situ grown FeSn films with a Helium lamp. I understand that reaching the high $T_N=370\text{K}$ is already challenging, but a weakness of this experiment remains that the highest temperature (385K), just 15K above T_N , does not give a large temperature window into the paramagnetic state. In fact, none of the reported changes are clearly happening at T_N (they start at 100K) and the signal is lost above 320K for the Dirac cone in Fig. 2(l-o).

The main claims of the paper are :

- (1)- The identification of the exchange-split kagome flat bands that persists above the magnetic transition, suggestive of a localized moment magnetism.
- (2)- An orbital selective renormalization of the $d_{xy}-d_{x^2-y^2}$ majority spin bands.

These 2 findings would be interesting, but I do not think they are reliably established for the following reasons.

(1)– The authors observe a broad peak around -1.2eV and, as they say « this features matches the location of the kagome flat band ». I agree with this, but there are many bands overlapping in this region (it is clear in Fig. 2d), from both minority and majority channels, and it is always difficult to detect unambiguously the dispersion of such high binding energy features. The « shift » with temperature could also be a broadening (the shape clearly changes), it is not particularly related to the transition as it starts at 100K, it could well depend on small structural changes with temperature (such as lattice expansion). I do not mean that it is impossible that it reflects some evolution of the flat band, but I do not think it can be used as an unambiguous proof of the exchange splitting, and even less of its evolution with temperature. This strongly diminishes the impact of the findings.

(2)– The claim of an orbital selective renormalization is mainly based on the small feature at the Fermi level in Fig. 3a that has no counterpart in the DFT calculation. Therefore the authors suggest it is the dx_{y^2} majority band renormalized by factor 13, which is indeed a spectacular change compared to the factor less than 2 usually observed in these systems ! However, the evidence for this assignment (« identical k_F ») is very weak, there could well be a band shifted from above E_F compared to the DFT or a surface state. Even worse, the authors want to assign the Dirac at -0.1eV in Fig. 4c to a dirac originally at -2eV in the calculation (Fig. 4e), but the dispersion of this cone is steep and certainly not renormalized by a factor 10, which makes this claim strongly inconsistent. Again, surface effects are not even discussed, whereas almost all of the previous studies (41,42) were devoted to the discussion of this feature as a surface Dirac cone. This could be discussed but not ignored as it is done here.

To conclude, I find the subject, and probably the data, interesting, but the discussion inappropriate. I think this manuscript should really not be published in the present form.

Reviewer #2 (Remarks to the Author):

The kagome lattice in FeSn is expected to host topological flat bands. Below a magnetic transition temperature $T_N = 370\text{ K}$, FeSn exhibits an AFM ground state. DFT calculations reveal that in the absence of local moments, the flat bands reside at E_F in the PM state, but they deviate from E_F due to exchange splitting in the AFM state. If the splitting disappears in the PM state, the magnetic ordering originates from Stoner instability. Otherwise, it originates from the long-range ordering of local moments. Previous ARPES experiments were conducted in the AFM state due to the high transition temperature, revealing flat bands located away from E_F . The authors performed temperature-dependent ARPES measurements on FeSn thin films across T_N . They observe that the exchange splitting persists above T_N , indicating that the magnetism originates from the presence of local moments rather than Stoner instability. I agree with this argument.

Another aspect addressed in this study is spin-orbit selective band renormalization. They observe narrow bands near E_F at Γ . Strong renormalization of the DFT bands is required to match the experimental data. The observed band crossing at -0.1 eV at K lacks a corresponding feature in the calculated bands. These discrepancies are attributed to spin-orbital selective band renormalization. Aside from those mentioned above, some noticeable discrepancies exist between experimental and calculated results. For instance, the hole-like band along K - Γ - K in Fig. 4b,d lacks a corresponding feature in the experimental data. FeSn has two types of charge-polarized

terminations, leading to substantial changes in the band structures. The authors only performed band calculations for bulk states. Furthermore, the presence of local moments suggests strong electron correlation in FeSn. DFT calculations may not provide an adequate start point for comprehending the electronic structure of FeSn. The evidence supporting selective band renormalization is insufficient when relying solely on the comparison between ARPES data and DFT bands of bulk states.

Minor comment:

In Fig. 1i, both spin-up and spin-down arrows are depicted on a single band at $T > T_c$. It appears as though each band can accommodate two electrons in the PM state within the local moment framework. This schematic representation may lead to confusion and misinterpretation.

Reviewer #1 (Remarks to the Author):

FeSn is a magnetic kagome compound, which raises important questions about the coexistence of magnetism, correlations and topology. This manuscript reports an ARPES study of its electronic structure across the magnetic transitions at $T_N=370\text{K}$. Although there have already been a number of ARPES studies of this system (41,42, but Moore et al. PRB 106, 115141 (2022) and Multer Communication Materials 4, 17 (2023) should also be quoted), it was never up to the magnetic transition. Understanding how the electronic structure evolves across the transition is an important issue as it should characterize the type of electronic correlations.

Reply:

We thank the Reviewer for the assessment that our experimental results, which reveal the evolution of the electronic structure of FeSn across the magnetic transition, is important. We are also delighted that the Reviewer appreciates the novelty of our work that we conducted the first ARPES measurement on FeSn across its high Neel temperature.

We are happy to follow the Reviewer's expert advice to add the other ARPES studies on FeSn into the references as Ref. 45 and Ref. 46.

The ARPES experiment is performed on in-situ grown FeSn films with a Helium lamp. I understand that reaching the high $T_N=370\text{K}$ is already challenging, but a weakness of this experiment remains that the highest temperature (385K), just 15K above T_N , does not give a large temperature window into the paramagnetic state. In fact, none of the reported changes are clearly happening at T_N (they start at 100K) and the signal is lost above 320K for the Dirac cone in Fig. 2(l-o).

Reply:

We are again thankful to the Reviewer for acknowledging the challenge that our work has overcome by reaching 370 K. The Reviewer is correct that the highest temperature reached in our experiments, 385 K, is not deep into the paramagnetic phase of FeSn. However, we would like to emphasize that two key observations are sufficient for making a conclusion on the nature of the magnetism: the full magnitude of the exchange splitting deep in the ordered phase and the portion of the splitting that disappears at the transition temperature. In a Stoner-type itinerant ferromagnet case, the spin splitting would vanish at the ordering temperature in an order parameter fashion. In a local moment case, the spin splitting would remain well above the ordering temperature. An example is that of Fe film, as shown below in Fig. R1, where by the Curie temperature, the measured spin splitting has closed by $\sim 50\%$ while truly disappearing at a higher temperature. Hence already measuring the spin splitting up through the ordering temperature is very informative of the nature of the magnetism, as the key is to measure near the transition. In our case, the large portion of the remaining exchange splitting at the ordering temperature indicates the local-moment nature of the magnetism, as can be compared to the Fe film case below. Although one could suppose that much deeper into the paramagnetic phase, the remaining exchange splitting would diminish as the local moments experience more thermal fluctuations, the nature of the magnetic order should already be revealed at the ordering temperature.

Such a means to determine the nature of magnetism has been widely employed in ARPES studies on other

er

magnetic materials, for example, on Fe_3GeTe_2 (Xu et al., PRB 101, 201104(R) (2020), now cited as Ref. 31). Regarding the Reviewer's comment that "none of the reported changes are clearly happening at T_N ", we would like to mention that for a continuous phase transition, the order parameter does not necessarily change sharply at the ordering temperature. However, it gradually increases on cooling. In fact, the tracking of the EDC peaks does show a larger rate of change closer to T_N and a slower change at lower temperatures (Fig. R2).

Lastly, we would like to mention that reaching 385 K is already at the limit of our ARPES system. It requires significant modification of our system to attempt higher temperatures, which is beyond the scope of this work. Nevertheless, we appreciate the challenge and motivation from the Reviewer and hope to expand our capabilities in the future.

Fig. R2 Same figure panels as Fig. 2g,m, with grey lines as a guide for the faster band shift closer to T_N .

The main claims of the paper are :

- (1)- The identification of the exchange-split kagome flat bands that persists above the magnetic transition, suggestive of a localized moment magnetism.
- (2)- An orbital selective renormalization of the d_{xy} - $d_{x^2-y^2}$ majority spin bands.

These 2 findings would be interesting, but I do not think they are reliably established for the following reasons.

Reply:

We thank the Reviewer for the concise summary of our work, and the assessment that our findings are interesting. We respond to the Reviewer's comments point-by-point below.

(1)– The authors observe a broad peak around -1.2eV and, as they say « this features matches the location of the kagome flat band ». I agree with this, but there are many bands overlapping in this region (it is clear in Fig. 2d), from both minority and majority channels, and it is always difficult to detect unambiguously the dispersion of such high binding energy features. The « shift » with temperature could also be a broadening (the shape clearly changes), it is not particularly related to the transition as it starts at 100K, it could well depend on small structural changes with temperature (such as lattice expansion). I do not mean that it is impossible that it reflects some evolution of the flat band, but I do not think it can be used as an unambiguous proof of the exchange splitting, and even less of its evolution with temperature. This strongly diminishes the impact of the findings.

Reply:

We thank the Reviewer for agreeing with us on the interpretation of the peak at -1.2 eV. The Reviewer is absolutely correct that there are many bands overlapping in this region. However, it is clear that the spin majority bands are dominant in this region (Fig. 2c), which is consistent with the fact that the peak at -1.2 eV shifts towards lower binding energy on warming. Moreover, the densely placed spin majority bands also agree with the strong intensity and the broadness of the peak at -1.2 eV.

The Reviewer raised a few excellent points regarding the validity of the band shift as a function of temperature, and whether our data reflects the exchange splitting. We reply to each of them as follows.

Fig. R3 Comparison of ARPES data with DFT calculations in PM and AFM phases.

First, the fact that our ARPES measured band structure matches the DFT calculations in the AFM phase better, but deviates from the DFT calculations in the PM phase (Fig. R3), serves as direct evidence for the exchange splitting and its persistence across T_N . Importantly, although we extracted the band shifts with temperature by carefully analyzing the high-quality data, the shifts are extremely small compared with the shifts anticipated in the DFT PM/AFM calculations. Therefore, the overwhelmingly persistent exchange splitting above T_N suggests a local-moment scenario, regardless the nature of the tiny band shifts. Hence, our main conclusion (1) remains intact.

We address the two other possibilities the Reviewer raised below.

The Reviewer is correct that the EDC peaks broaden with increasing temperature, as one would expect due to thermal effects. However, the shape of the profile does not change, as they can be fitted by one (away from Γ) or two (closer to Γ) Lorentzian functions and a linear background at all temperatures (Fig. S3). To check whether the shift of the peak position is meaningful, one could compare the shift of peak position with the change of the full width at half maximum (FWHM) of the Lorentzian fit. If the former is comparable to the latter, it is reasonable to deem the peak shift a real shift, instead of an illusion due to broadening. In Fig. R4, we take the EDC near Γ as an example (corresponding to Fig. S3f), and plot the peak position and FWHM of the right Lorentzian fit as a function of temperature. From 82 K to 385 K, the peak position shifts by 0.19 eV, while the FWHM increases by 0.2 eV, comparable to the peak shift. Therefore, the EDC peak shift, although much smaller compared with the exchange splitting, is valid. In addition, the small shift across T_N and the lack of strong temperature dependence is precisely consistent with the persistent local moment understanding of the magnetism.

Fig. R4 EDC Lorentzian fit, peak position and FWHM as a function of the temperature.

Regarding band shifts as a result of lattice change with temperature, a previous study found an in-plane lattice expansion of 0.4% over a 300 K temperature window (Sales et al., Phys. Rev. Mater. 3, 114203 (2019)). To explore the change of band structure due to the change of lattice parameters, we performed additional DFT calculations as a function of strain. Fig. R5 (now Fig. S6) shows the comparison of the DFT bands with in-plane zero (blue) and 0.5% tensile strain (red). Although discernible changes exist, most of the bands shift towards lower binding energy with tensile strain. In particular, the band bottom at Γ and the top of the Dirac bands (green boxes in Fig. R5) shift in the same direction, contradicting the temperature-dependent ARPES results which is associated with the spin majority or spin minority character of the bands. Furthermore, the predicted shift at the band bottom near Γ with 0.5% tensile strain is ~ 0.04 eV, whereas the ARPES data shows a 0.2 eV shift from 82 K to 385 K. Therefore, band shifts induced by thermal lattice expansion cannot account for the total amount of the shifts observed nor the direction of the shift observed.

Fig. R5 (also Fig. S6) DFT calculations as a function of strain.

We gratefully acknowledge the Reviewer's advice on a more careful interpretation of the ARPES results. We now added the Supplementary Note 2 to discuss the impact of lattice expansion on the band structure.

(2)– The claim of an orbital selective renormalization is mainly based on the small feature at the Fermi level in Fig. 3a that has no counterpart in the DFT calculation. Therefore the authors suggest it is the

$d_{xy}/d_{x^2-y^2}$ majority band renormalized by factor 13, which is indeed a spectacular change compared to the factor less than 2 usually observed in these systems!

Reply:

We thank the Reviewer for acknowledging the significance of our findings by calling the observed band renormalization “a spectacular change”. The Reviewer is perfectly correct that most kagome metals studied thus far show modest correlation effects, which makes our discoveries here even more remarkable. It is interesting to note that in some Fe-based superconductors, for example $\text{FeTe}_x\text{Se}_{1-x}$, a large renormalization factor of ~ 20 has been seen in the bands of the d_{xy} orbital character (Yi et al., npj Quant. Mater. 2, 57, (2017), Huang et al., Comm. Phys. 5, 29 (2022)), which draws an interesting connection to our work, considering that Fe 3d orbitals play a crucial role in both cases.

However, the evidence for this assignment (« identical k_F ») is very weak, there could well be a band shifted from above E_F compared to the DFT or a surface state. Even worse, the authors want to assign the Dirac at -0.1eV in Fig. 4c to a dirac originally at -2eV in the calculation (Fig. 4e), but the dispersion of this cone is steep and certainly not renormalized by a factor 10, which makes this claim strongly inconsistent. Again, surface effects are not even discussed, whereas almost all of the previous studies (41,42) were devoted to the discussion of this feature as a surface Dirac cone. This could be discussed but not ignored as it is done here.

Reply:

We thank the Reviewer for pointing out the other possibilities as the origin of the shallow bands near E_F . We agree with the Reviewer that “identical k_F ” alone should not be a sufficient reason for assigning the bands. However, we would like to emphasize that the overall ARPES band structure shows excellent agreement with the DFT calculations. Therefore, it is unlikely that only the shallow bands are shifted down by hundreds of meV from above E_F (Fig. R6 greens arrows). Furthermore, the large effective mass associated with the shallow bands ($\sim 7m_e$) still cannot be accounted for with this interpretation. Therefore, it is unlikely that the shallow bands are some bands above E_F that are shifted down in energy. There exist no bands from the DFT calculations with this narrow bandwidth.

Fig. R6 Possibility on empty states shifted down.

To strengthen the assignment of the narrow bands to the electron-like bands near the Γ point, we additionally performed the SPR-KKR band structure calculations based on the Green’s functions, complementary to the VASP DFT calculations provided previously. Quite interestingly, while the majority of the bands, such as the Dirac cones at K, show good consistency between these two kinds of calculations, the electron-like bands at Γ (the same ones as the renormalized bands in ARPES) are renormalized by a factor of ~ 2 in SPR-KKR compared with VASP DFT calculations (Fig. R7, now Fig. S8). While it is unclear why SPR-KKR predicts that these electron-like bands should be selectively renormalized, it points to the origin of the very narrow electron-like bands near Γ in the ARPES results, that they are further renormalized from the calculated electron-like bands at Γ . Furthermore, we also performed DMFT calculations which exhibit an overall renormalization effect (Fig. R7). Interestingly, the electron-like bands at Γ are again subject to the strongest renormalization effect, consistent with SPR-KKR calculations and ARPES observations.

Figure R7 (also Fig. S8) Comparison of SPR-KKR, VASP and DMFT calculations. Selectively renormalized bands are enclosed in the green boxes.

Another piece of evidence supporting the assignment of the narrow bands is the remarkable resemblance of the constant energy contours (CECs) of ARPES data and both types of calculations at $k_z = \pi$, which show the flower-shaped feature (Fig. R8, or revised Fig. 4, g-i). As it goes to higher binding energy, the flower petals shrink in size and maintain a hexagonal shape, which is consistent between experiment and calculations when the renormalization factor is considered (Fig. R8 g-i). Figure R8 a-f facilitate a clearer comparison between ARPES, SPR-KKR and VASP. This indicates that not only does this band match along the high symmetry direction, but the entire momentum-dependence across the Brillouin zone matches this feature as well. While the Dirac cones ($d_{xy} + d_{x^2-y^2}$ spin minority) show good match between all three panels (Fig. R8 d-f), it is evident that the electron-like bands near E_F at Γ ($d_{xy} + d_{x^2-y^2}$ spin majority) are strongly renormalized from calculations, while the renormalization factors are different from SPR-KKR and VASP (Fig. R8 a-c). Notably, the hole-like band at Γ ($d_{xz} + d_{yz}$ spin minority, see Fig. 3d) shows a shift in energy, but a negligible renormalization factor (Fig. R8 a-c), further supporting the argument that the renormalization is extremely enhanced only in the $d_{xy} + d_{x^2-y^2}$ spin majority spin-orbital channel. The rapid depletion of the spectral weight of the strongly renormalized electron-like bands over warming further suggest a possible relation with the orbital-selective Mott transition, similar to that observed in the iron-based superconductors (Fig. R8 j-l). Lastly, we show a direct comparison of the renormalization factors in different spin-orbital channels in order to better convey our argument (Fig. R8m). We substantively revised Fig. 4 in the manuscript to include these changes.

Figure R8 (revised Fig. 4) Strong band renormalization in $d_{xy} + d_{x^2-y^2}$ spin majority channel (full caption in revised manuscript)

The Reviewer also raised an important issue that the shallow bands could be surface states. To rule out this possibility, we performed additional calculations and experiments as follows.

First, we performed slab-DFT calculations to account for the surface states. Fig. R9 (now Fig. S9) shows the surface states on the Sn termination and the kagome termination. The kagome surface states deviate significantly from the ARPES results. The Sn surface calculation shows some resemblance with the bulk DFT calculation, but it does not exhibit the extremely narrow electron-like bands near E_F at Γ as seen in ARPES. Similar magnitude of renormalization is still required to adapt the Sn surface electron-like bands to the experimental ones. Therefore, one could not deem the surface states a more likely origin for the strongly renormalized bands than the bulk states. We added Supplementary Note 3 to discuss surface DFT calculations.

Figure R9 (now Fig. S9) Surface DFT calculations

Second, it is extremely unlikely that the strongly renormalized bands are surface states because of their remarkable robustness against various perturbations. As we show in Fig. R10 (revised Fig. S10), these bands can survive thermal cycling from 81 K to 300 K and back to 110 K, which is unlike typical trivial surface states (for example, see Ref. 45). We also tried capping the as-grown FeSn films with amorphous Se and subsequently taking them to a synchrotron ARPES beamline and removing the capping layer by annealing. As shown in Fig. R10, while the synchrotron ARPES data quality on the samples that went through this brutal process is expectedly diminished, the strongly renormalized flower-shaped feature is still clearly observed. It is highly unlikely that surface states can survive this process because of the additional deposition on the surface. In fact, slight deposition on a cleaved surface has been purposely employed to destroy the surface states and unravel the bulk states in ARPES experiments (Cheng et al., PRB 109, 075150 (2024), now Ref. 47). We added Supplementary Note 4 to discuss the robustness of the strongly renormalized bands.

Figure R10 (revised Fig. S10) Robustness of the strongly-renormalized bands. **a,b** Before and after thermal cycling up to room temperature. **c,d** Fermi surface map and high symmetry cut after capping and decapping the FeSn film with Se. Red arrows mark the persistent strongly-renormalized flower feature.

Third, the strongly renormalized electron-like bands at Γ can be observed on both the kagome termination and the Sn termination, which directly rules out these bands being surface states. The additional experiments were performed using synchrotron ARPES with a 20-micron-sized beam spot, which enables resolving the two terminations. Because the FeSn films have a homogeneous Sn termination, we cleaved FeSn single crystals and observed both terminations based on the distinct Sn core level profiles (Fig. R11 b,e, also Fig. S11) (see Ref. 43). The Fermi surface map on both terminations show an electron pocket at Γ , which has a very narrow bandwidth (Fig. R11 c,d,f,g). Although they do not exhibit the flower shape, the narrow bandwidth and large effective mass suggest a similar origin as the flower-shaped bands observed in the films. Importantly, this strongly renormalized electron-like band appears on both the Sn and the kagome terminations, indicating that it is not a surface state. We added Supplementary Note 5 to discuss the termination-independence of the strongly renormalized bands.

Lastly, regarding the second band crossing at K which we originally assigned to the renormalized spin majority Dirac cone, we now agree with the Reviewer that it is probably a surface state for the following reasons: (1) Ref. 45 demonstrates that this band crossing disappears after thermal cycling; (2) in our thin film data where surface states are mostly quenched, this band crossing can hardly be seen (Fig. 2j) and we had to resolve it in the second derivative; (3) we also observed this crossing in our single crystal data and it only exists on the Sn termination but not on the kagome termination (Fig. R11 d,g). Since this feature

has been extensively studied in Ref. 45, we now removed it from our manuscript. We are thankful to the Reviewer for correcting our interpretation of the data.

Figure R11 (also Fig. S11) Termination dependence of the strongly renormalized bands (red arrows in c,d and f,g). Yellow arrow in d points to the surface band crossing. Full caption in Fig. S11.

To conclude, I find the subject, and probably the data, interesting, but the discussion inappropriate. I think this manuscript should really not be published in the present form.

Reply:

We sincerely thank the Reviewer again for finding the subject and our data interesting, and for providing so much expert advice on how to improve the manuscript. With the substantive revision (additional experiments and calculations, majorly revised Fig. 4 and five additional supplementary figures), we are confident that our manuscript has been significantly improved and meets the criteria for publication.

 Reviewer #2 (Remarks to the Author):

The kagome lattice in FeSn is expected to host topological flat bands. Below a magnetic transition temperature $T_N = 370$ K, FeSn exhibits an AFM ground state. DFT calculations reveal that in the absence of local moments, the flat bands reside at E_F in the PM state, but they deviate from E_F due to exchange splitting in the AFM state. If the splitting disappears in the PM state, the magnetic ordering originates from Stoner instability. Otherwise, it originates from the long-range ordering of local moments. Previous ARPES experiments were conducted in the AFM state due to the high transition temperature, revealing flat bands located away from E_F . The authors performed temperature-dependent ARPES measurements

on FeSn thin films across TN. They observe that the exchange splitting persists above TN, indicating that the magnetism originates from the presence of local moments rather than Stoner instability. I agree with this argument.

Reply:

We thank the Reviewer for the concise summary of our work and for pointing out our accomplishment that we performed ARPES measurement on FeSn across its high transition temperature for the first time. We appreciate the Reviewer agreeing with our argument.

Another aspect addressed in this study is spin-orbit selective band renormalization. They observe narrow bands near EF at Gamma. Strong renormalization of the DFT bands is required to match the experimental data. The observed band crossing at -0.1 eV at K lacks a corresponding feature in the calculated bands. These discrepancies are attributed to spin-orbital selective band renormalization.

Aside from those mentioned above, some noticeable discrepancies exist between experimental and calculated results. For instance, the hole-like band along K-Gamma-K in Fig. 4b,d lacks a corresponding feature in the experimental data.

Reply:

We thank the Reviewer again for the nice summary of the second part of our results. As in our response to the first Reviewer's questions and the in the revised Fig. 4 (Fig. R8 below), we further performed SPR-KKR calculations, complementary to the VASP DFT calculations (Fig. R7). An interesting finding is that from VASP to SPR-KKR to ARPES, the flower-shaped electron pockets at Γ show a progressively stronger renormalization effect (Fig. R8 a-c). In contrast, the majority of the other bands, for example the Dirac cones, are consistent between ARPES and two calculations (Fig. R8 d-f). The hole-like band at Γ that the Reviewer mentioned can be seen in Fig. R8 a-c, which shifts to lower binding energy as the electron-like bands get renormalized. Notably, the hole-like band does not exhibit apparent renormalization effect, which is consistent with its spin-orbital character being $d_{xz} + d_{yz}$ spin minority (Fig. 3d). Because the strong renormalization only affects the $d_{xy} + d_{x^2-y^2}$ spin majority channel (Fig. R8m), the hole-like band is almost not renormalized. Therefore, our argument is self-consistent. Furthermore, the flower-shaped pockets, constituted by the renormalized electron-like bands and the unrenormalized hole-like bands, show remarkable consistency between ARPES data and SPR-KKR and VASP calculations (Fig. R8 g-i) after the renormalization effect is considered. This further provides strong evidence for matching the ARPES data with calculations.

Figure R8 (revised Fig. 4) Strong band renormalization in $d_{xy} + d_{x^2-y^2}$ spin majority channel (full caption in revised manuscript)

FeSn has two types of charge-polarized terminations, leading to substantial changes in the band structures. The authors only performed band calculations for bulk states.

Reply:

We thank the Reviewer for raising the possibility of surface states. To rule this possibility out, we additionally performed synchrotron ARPES measurements on capped and de-capped FeSn films as well as cleaved FeSn single crystals, complemented by DFT slab calculations for surface states. Reviewer #1 asked a similar question and our response can be found there. For the convenience of the Reviewer, we duplicated our response here:

First, we performed slab-DFT calculations to account for the surface states. Fig. R9 (now Fig. S9) shows the surface states on the Sn termination and the kagome termination. The kagome surface states deviate significantly from the ARPES results. The Sn surface calculation shows some resemblance with the bulk DFT calculation, but it does not exhibit the extremely narrow electron-like bands near E_F at Γ as seen in ARPES. Similar magnitude of renormalization is still required to adapt the Sn surface electron-like bands to the experimental ones. Therefore, one could not deem the surface states a more likely origin for the strongly renormalized bands than the bulk states. We added Supplementary Note 3 to discuss surface DFT calculations.

Figure R9 (now Fig. S9) Surface DFT calculations

Second, it is extremely unlikely that the strongly renormalized bands are surface states because of their remarkable robustness against various perturbations. As we show in Fig. R10 (revised Fig. S10), these bands can survive thermal cycling from 81 K to 300 K and back to 110 K, which is unlike typical surface states (for example, see Ref. 45). We also tried capping the as-grown FeSn films with amorphous Se and subsequently taking them to a synchrotron ARPES beamline and removing the capping layer by annealing. As shown in Fig. R10, while the synchrotron ARPES data quality on the samples that went through this brutal process is expectedly diminished, the strongly renormalized flower-shaped feature is still clearly observed. It is extremely unlikely that surface states can survive this process because of the additional deposition on the surface. In fact, slight deposition on a cleaved surface has been purposely employed to destroy the surface states and unravel the bulk states in ARPES experiments (Cheng et al., PRB 109, 075150 (2024), now Ref. 47). We added Supplementary Note 4 to discuss the robustness of the strongly renormalized bands.

Figure R10 (revised Fig. S10) Robustness of the strongly-renormalized bands. **a,b** Before and after thermal cycling up to room temperature. **c,d** Fermi surface map and high symmetry cut after capping and decapping the FeSn film with Se. Red arrows mark the persistent strongly-renormalized flower feature.

Third, the strongly renormalized electron-like bands at Γ can be observed on both the kagome termination and the Sn termination, which directly rules out these bands being surface states. The additional experiments were performed using synchrotron ARPES with a 20-micron-sized beam spot, which enables resolving the two terminations. Because the FeSn films have a homogeneous Sn termination, we cleaved

FeSn single crystals and observed both terminations based on the distinct Sn core level profiles (Fig. R11 b,e, also Fig. S11) (see Ref. 43). The Fermi surface map on both terminations show an electron pocket at Γ , which has a very narrow bandwidth (Fig. R11 c,d,f,g). Although they do not exhibit the flower shape, the narrow bandwidth and large effective mass suggest a similar origin as the flower-shaped bands observed in the films. Importantly, this strongly renormalized electron-like band appears on both the Sn and the kagome terminations, indicating that it is not a surface state. We added Supplementary Note 5 to discuss the termination-independence of the strongly renormalized bands.

Lastly, regarding the second band crossing at K which we originally assigned to the renormalized spin majority Dirac cone, we now agree with the Reviewer that it is probably a surface state for the following reasons: (1) Ref. 45 demonstrates that this band crossing disappears after thermal cycling; (2) in our thin film data where surface states are mostly quenched, this band crossing can hardly be seen (Fig. 2j) and we had to resolve it in the second derivative; (3) we also observed this crossing in our single crystal data and it only exists on the Sn termination but not on the kagome termination (Fig. R11 d,g). Since this feature has been extensively studied in Ref. 45, we now removed it from our manuscript. We are thankful to the Reviewer for correcting our interpretation of the data.

Figure R11 (also Fig. S11) Termination dependence of the strongly renormalized bands (red arrows in c,d and f,g). Yellow arrow in d points to the surface band crossing. Full caption in Fig. S11.

Furthermore, the presence of local moments suggests strong electron correlation in FeSn. DFT calculations may not provide an adequate start point for comprehending the electronic structure of FeSn. The evidence supporting selective band renormalization is insufficient when relying solely on the comparison between ARPES data and DFT bands of bulk states.

Reply:

The Reviewer is perfectly correct that DFT may not reflect the realistic electronic structure when strong correlation is present. However, we would like to emphasize that our ARPES data overall shows an excellent agreement with DFT calculations, largely justifying the use of DFT in this case. A similar situation might be the case of Fe-based superconductors, where DFT calculations qualitatively match the ARPES band structure after an orbital-dependent renormalization is included (Yi et al., npj Quant. Mater. 2, 57, (2017)). It could be a similar situation here, considering that Fe 3d orbitals play an important role in both cases.

To complement the VASP DFT calculations, we further performed the SPR-KKR calculations based on Green's functions, and DMFT calculations. As shown in Fig. R7, interesting similarities and differences are observed and discussed above. The comparison between ARPES data, SPR-KKR calculations, VASP calculations and DMFT calculations further strengthens our argument on the spin-orbital selective renormalization effect in FeSn.

Minor comment:

In Fig. 1i, both spin-up and spin-down arrows are depicted on a single band at $T > T_c$. It appears as though each band can accommodate two electrons in the PM state within the local moment framework. This schematic representation may lead to confusion and misinterpretation.

Reply:

We thank the Reviewer for pointing out this issue. We modified the schematic as shown in Fig. R12 and added “Dashed lines indicate two degenerate cases of exchange splitting in different local moment regions” in Fig. 1i caption to avoid the possible confusion.

Figure R12 (revised Fig. 1i) Modified schematic for the exchange splitting in the local moment case.

To conclude, we sincerely thank both Reviewers again for their excellent advice that helps us improve the manuscript. With the additional experimental data and calculations, we substantively revised our manuscript with the changes summarized in the following list:

New measurements:

- (1) Synchrotron ARPES measurements on FeSn thin films enabled by Se capping and de-capping;
- (2) Synchrotron ARPES measurements on FeSn single crystals with termination dependence;

New calculations:

- (3) Slab DFT calculations for surface states;
- (4) Strain-dependence DFT calculations;
- (5) SPR-KKR calculations of FeSn band structure;
- (6) DMFT calculations of FeSn band structure and renormalization.

The above new information have been **incorporated** into the manuscript and supplementary materials as:

- (1) revised main Fig. 4 and corresponding text to include the comparison with the new SPR-KKR calculations.
- (2) Supplementary Note 2 and Fig. S6;
- (3) Fig. S8;
- (4) Supplementary Note 3 and Fig. S9;
- (5) Supplementary Note 4 and Fig. S10;
- (6) Supplementary Note 5 and Fig. S11.
- (7) Due to the new measurements and calculations carried out, we have added 8 new co-authors to the author list.
- (8) We added new references: 31, 32, 45-47, 62-68.

We hope that the manuscript now meets the criteria for publication in *Nature Communications*.

REVIEWER COMMENTS

Reviewer #1 (Remarks to the Author):

Unfortunately, I had major concerns with the claims in this manuscript and none of them were really considered for this resubmission (except for the one about the renormalized Dirac cone that has been removed). The authors provide new calculations and more information about surface states contribution, this is fine, but this does not strengthen much their point in my opinion.

I briefly summarize again my two objections :

(1) Temperature dependence : *If* the flat band was a clear and isolated feature, I would agree its temperature dependence would be a good sign of the nature of the transition. However, it is very ill-defined and as its temperature evolution is not clearly related to T_N , I do not trust any conclusion drawn out of this. The Dirac cone could not be measured up to the magnetic phase, so that it cannot be used for this discussion.

(2) Orbital selectivity : *If* we were in a case of a simple electronic structure, well described by DFT, I might agree it would be worth discussing the origin of the narrow electron bands at Gamma. However, to me, there is almost no clear agreement between the measurement and DFT, except for the Dirac cone. This is really far from the « excellent agreement » claimed in the answer.

Sincerely, if I look at the figure attached, what is the match for bands 1 and 2 ? Is it the $dxz+dyz$? If yes, why is the upper part of the cone so poorly described ? For the « VHS », the manuscript itself (page 5) recognizes the problem with the electron-like band 3 (not seen) and then does band 1 corresponds to the feature at -0.2 or -0.4eV or none ? I could show that adding the spin majority contribution does not help.

Now, for the bands at Gamma (left). One would guess from the right panels that the measurement is closer to $k_z=0$, as for $dxz+dyz$ all $k_z=1$ bands are really off, but then why is the narrow band at Gamma compared to $k_z=1$? Because there is a better match for k_F ? And what is the correspondence for bands 4 ? Why is 5 not seen or why is 6 seen if we are close to $k_z=0$?

I know very well that other photon energies, polarizations, etc., might clarify and complement the picture, but without a solid description of the band structure, I find it impossible to trust the conclusion on orbital selectivity.

I acknowledge again the value and interest of the work, but to me the conclusions are not supported at this stage and I advise against publication.

Reviewer #3 (Remarks to the Author):

The study of flat band-induced metallic magnetism has been a vibrant area of research ever since Mielke's pioneering theoretical prediction over 30 years ago of a ferromagnetic ground state in the Hubbard model on the Kagome lattice. This field has gained renewed attention with the recent discoveries of superconductivity and orbital magnetism in twisted bilayer graphene, as well as unconventional charge density waves and superconductivity in metallic bulk Kagome systems. Correlated flat bands are now recognized as one of the most exciting and timely topics in modern condensed matter physics.

In this context, the present work offers a significant contribution by providing the first detailed study of the temperature-dependent evolution of the electronic structure in the well-studied Kagome flat band magnet FeSn. The authors present compelling evidence that the magnetism in this system is primarily due to local moments and Heisenberg ferromagnetism, rather than itinerant electrons leading to Stoner-type ferromagnetism. This finding is likely to play a crucial role in shaping future theoretical models that aim to accurately describe metallic Kagome magnets, such as development of compact molecular orbitals that respect the topological constraints of the flat bands. It is clear that this research will be of great interest to a wide audience, and I strongly recommend its publication in Nature Communications.

That said, while I find the interpretation of the temperature evolution quite convincing, I do have some remaining concerns regarding the second major finding of the paper, specifically related to the interpretation of the flower-like bands near the \bar{G} points. This was already mentioned by the other referees, and I am not yet fully convinced by the response that the authors have provided.

The new analysis presented in Figures 4g–i provides strong arguments for interpreting the flower-like bands as renormalized bulk bands. However, a potential inconsistency arises when comparing the flower-like Fermi surface in Fig. 4g, which is attributed to the Sn termination, with the “Kagome” termination shown in Fig. S11. In the latter case, the six petals of the Fermi surface are absent. This behavior aligns with the termination-dependent flower-like Fermi surface reported by Kang et al. (Nat. Mater. 19, 163 (2020)), where the six petals appear only for the Sn-terminated surface (Fig. 2c). If these petals are indeed bulk states, their absence on the “Kagome termination” is puzzling. I suggest that the authors address this apparent inconsistency in the discussion section. Additionally, mentioning that photon energy-dependent measurements could be conducted in future studies to further explore the dimensionality (and thus potential surface character) of the flower-like Fermi surface would be valuable.

Furthermore, the strong renormalization observed in the bands near the Fermi level raises some questions. Is it really plausible to have such significant renormalization while still maintaining sharp bands near the Fermi level? A large renormalization typically implies a large real part of the self-energy, which, through the Kramers-Kronig relations, should correspond to a substantial imaginary part of the self-energy. However, the lifetime broadening here appears minimal. The authors might consider discussing the relationship between strong renormalization and lifetime broadening in other materials to clarify this point.

Beyond this primary concern, I have a few minor comments that the authors might find helpful:

1. The authors mention that their films are primarily Sn-terminated. It would be beneficial to provide the experimental evidence that supports this conclusion.
2. Have the authors studied the thicker films used for magnetization measurements with ARPES? If not, do they anticipate any differences in the electronic structure between thin and thick films?
3. In Fig. 2a, the peak in the DOS near E_F in the PM state is not symmetrically split around the Fermi level in the AFM phase. Is there a straightforward explanation for this?
4. The sentence, "Although most bands in the DFT calculations have a good match in the ARPES data (Fig. 2d,j), the calculated spin majority electron pocket along $K_{\text{bar}}-G_{\text{bar}}-K_{\text{bar}}$ within 0.5 eV below E_F seems to have no experimental counterpart (Fig. 2d)," is somewhat unclear. I don't see an electron pocket in Fig. 2d.
5. Do the authors have any insights into why the bands corresponding to the vHS below E_F are not observed?
6. In Fig. 3a, the green boxes appear quite large. Perhaps arrows pointing to the narrow bands at Gamma would be more precise.
7. Is it known why strongly renormalized bands might exhibit a more pronounced temperature dependence in their coherence factor compared to more weakly renormalized bands? If so, a brief comment on this would be appreciated.

I hope that these comments are helpful and constructive, as my intention is to further improve the already impressive quality of the paper. I look forward to seeing the final version of this exciting work.

Reviewer #1 (Remarks to the Author):

Unfortunately, I had major concerns with the claims in this manuscript and none of them were really considered for this resubmission (except for the one about the renormalized Dirac cone that has been removed). The authors provide new calculations and more information about surface states contribution, this is fine, but this does not strengthen much their point in my opinion.

Reply: We thank the Reviewer for their time and effort in reviewing our manuscript. We respectfully disagree that none of the Reviewer's concerns were considered in our last revision. As can be referred to in our previous response letter, here is a brief recap of the points raised by the Reviewer from the last round and our key response to each:

- Needing a large window into the paramagnetic state:
 - We have illustrated that we do not need a large window into the paramagnetic state by referring to the Fe film result from literature. The portion of exchange splitting closed across T_N is sufficient to distinguish the two scenarios. Both Reviewers 2 and 3 agree with this conclusion. This method is routinely done, such as for Fe_3GeTe_2 and Cr_2Te_3 (see Fig. R3 below) as recent examples.
- Whether the shift of the flat band is due to broadening:
 - We have provided fitting results on the peak width and position to specifically exclude this possibility.
- Whether the shift of the bands could be due to thermal expansion:
 - We have provided new calculations examining the effect of lattice expansion and have excluded this possibility.
- Assignment of the renormalized band:
 - We have provided new SPR-KKR calculations, which together with VASP calculations of constant energy contours show excellent momentum-dependence with the observed dispersion, after taking into consideration the renormalization of the energy scale.
- Whether the renormalized band could be a surface state:
 - We have shown that this is not a surface state by: i) carrying out capping/de-capping synchrotron measurements of the film sample; ii) termination-independence of this feature from measurement of a single crystal sample; iii) termination-dependent slab calculations that do not show such renormalized feature.

We sincerely thank the Reviewer again for raising these points that helped further improve our manuscript. As we addressed each of these points by new measurements and calculations, we believe that we have significantly strengthened our conclusions. As the Reviewer reiterates some of these points here, we clarify them point-by-point as follows.

I briefly summarize again my two objections:

(1) Temperature dependence: *If* the flat band was a clear and isolated feature, I would agree its temperature dependence would be a good sign of the nature of the transition. However, it is very ill-

defined and as its temperature evolution is not clearly related to T_N , I do not trust any conclusion drawn out of this. The Dirac cone could not be measured up to the magnetic phase, so that it cannot be used for this discussion.

Reply: We can answer this by considering both large and small energy scales. In Fig. R1 below (reproduced from our supplementary Fig. S8), we show the temperature dependence of the large energy scale spectra. The group of flat bands pointed by the blue arrow can be clearly observed for both cuts at all temperatures, including $380\text{ K} > T_N$. Even though there exists thermal broadening due to the high temperature, even in raw data we can see that the dispersions are closer to the DFT calculations for the ordered phase with the large exchange splitting. The general lineshape of the EDCs do not show remarkable change across T_N , indicating little change in the exchange splitting. We further present the second derivative of these cuts in Fig. R2 for better visualization of the flat band. While we agree that the flat band in FeSn is not isolated from other bands, we emphasize that the exchange splitting applies to not only the flat band, but many other bands near E_F , as illustrated in our DFT calculations. In our manuscript, we utilize the opposite, but *small*, shifts of the bands that belong to the spin majority and spin minority categories, to demonstrate the persistent exchange splitting up to T_N . We do not solely rely on the shift of the flat band at higher binding energy to draw the conclusion (although we also show that the same behavior applies to the flat band above), but also examine other important bands, including the quadratic band bottom and the Dirac cone. A consistent picture is clear from the *small* shifts of all the bands, that is, the exchange splitting of the spin majority and spin minority bands is persistent across T_N , suggesting a local-moment driven magnetism in FeSn.

Figure R1 (reproduced from Fig. S8) **Temperature dependence of the spin majority flat band.** $\bar{K} - \bar{M} - \bar{K}$ and $\bar{K} - \bar{\Gamma} - \bar{K}$ cuts taken at 83 K, 230 K and 380 K and their momentum-integrated EDCs.

Figure R2 Second derivative of the $\bar{K} - \bar{M} - \bar{K}$ and $\bar{K} - \bar{\Gamma} - \bar{K}$ cuts taken at 83 K, 230 K and 380 K.

Again, we emphasize that the bands *not* having a strong response at T_N is in itself the key observation indicative of the local-moment picture. This is acknowledged by Reviewers 2 and 3, as well as well-accepted in the literature, such as shown in Fig. R3, reproduced from Zhong et al., Nat. Comm. 14, 5340 (2023). Specifically, the energy shift of the 1ML case shows no response at T_N . This is indeed taken as evidence that it is local-moment (Heisenberg-type) ferromagnetism.

Figure R3 Stoner-type and Heisenberg-type ferromagnetism in thin Cr_2Te_3 (reproduced from Zhong et al., Nat. Comm. 14, 5340 (2023)).

We thank the Reviewer for stressing that we should measure the Dirac cone across T_N . Here, we provide all raw data of the Dirac bands from base temperature to *above* T_N (up to 386 K) in Fig. R4 (now Fig. S7). The Dirac cones can be clearly observed over the entire temperature window. Due to the thermal broadening at high temperatures as well as the large velocity of the Dirac bands that smears out the energy distribution curves (EDCs), it is better to examine the momentum distribution curves (MDCs) to track the shift (as indicated by the green line in Fig. R4 a). Its temperature evolution in panels b and c shows that the Dirac bands can be tracked over the entire temperature window. The shift of the peak is consistent with a downward shift as temperature is increased (panel e)—consistent with its spin minority nature as we have discussed in the manuscript. Panel d provides the fitting of the peaks as a function of temperature.

We thank the Reviewer again for encouraging us to devise a better way to extract the shift of the Dirac bands up to 386 K, which is above T_N . We have added Fig. R4 into the Supplementary Information as the new Fig. S7, which is referenced in the revised text. We also replaced Fig. 2k with the same cut taken at 386 K $>$ T_N .

Figure R4 (now Fig. S7) Temperature evolution of the Dirac bands and the fitting of the MDCs.

(2) Orbital selectivity: *If* we were in a case of a simple electronic structure, well described by DFT, I might agree it would be worth discussing the origin of the narrow electron bands at Gamma. However, to me, there is almost no clear agreement between the measurement and DFT, except for the Dirac cone. This is really far from the « excellent agreement » claimed in the answer.

Sincerely, if I look at the figure attached, what is the match for bands 1 and 2? Is it the $dxz+dyz$? If yes, why is the upper part of the cone so poorly described? For the « VHS », the manuscript itself (page 5) recognizes the problem with the electron-like band 3 (not seen) and then does band 1 corresponds to the feature at -0.2 or -0.4eV or none? I could show that adding the spin majority contribution does not help.

Now, for the bands at Gamma (left). One would guess from the right panels that the measurement is closer to $k_z=0$, as for $dxz+dyz$ all $k_z=1$ bands are really off, but then why is the narrow band at Gamma compared to $k_z=1$? Because there is a better match for k_F ? And what is the correspondence for bands 4? Why is 5 not seen or why is 6 seen if we are close to $k_z=0$?

I know very well that other photon energies, polarizations, etc., might clarify and complement the picture, but without a solid description of the band structure, I find it impossible to trust the conclusion on orbital selectivity.

Reply: First of all, we sincerely thank the Reviewer for examining our experimental and calculated band structure in detail, which helps us rule out all other possibilities and further strengthen our conclusions. A one-to-one mapping between measured bands and the bands calculated with DFT would be desired in an ideal world. However, for most multi-orbital and multi-band materials this is rarely the case, especially for d-electron systems where multi-orbitals and strong correlations tremendously complicate the problem. In practice nonetheless, this has not prevented the community from addressing and advancing on the intriguing point of orbital-selective or band-selective correlation effects. A recent example is the work on Fe_3Sn_2 that reports anomalous electrons (Ekahana et al., *Nature* 627, 67 (2024)), where the match with overall valence band structure is not discussed but the focus is solely on the electron bands at the $\bar{\Gamma}$ point alone. As we are sure that as an expert in the ARPES community, the Reviewer must also appreciate that the aspect of orbital-selective correlation effect is an important one for the theoretical treatment of moderately correlated electron systems, which is a key challenge in modern condensed matter physics as it sits in between the two well-developed limits of single-particle DFT and Mott physics (Georges and Kotliar, *Physics Today* 77, 46 (2024)). In recent years, we have also seen theoretical models built on such scheme to treat topological flat bands in moiré systems where the coexistence of local and delocalized electronic states is mapped to different regions in momentum space (Song and Bernevig, *Phys. Rev. Lett.* 129, 047601 (2022); Datta et al., *Nat. Comm.* 14, 5036 (2023)). In all of these systems, two energy scales are important—the overall renormalization of the valence bandwidth and the much stronger renormalization of certain portions of bands associated with a selective degree of freedom (orbital, band, momentum, valley, spin). When these two renormalization factors are starkly distinct, the system exhibits a selective correlation effect. In the case of FeSn , we have examined the overall bandwidth of the Fe 3d orbitals. Both the location of the kagome flat bands (representing the energy scale of the exchange splitting) as well as the dispersive Dirac states clearly exhibit renormalization factors on the scale close to 1. In stark contrast, the bandwidth of the electron bands at the $\bar{\Gamma}$ point is clearly much smaller. Hence the key to demonstrating orbital-selective correlation effect is to demonstrate that the renormalized electron band that we observe is indeed the wider electron bands that we compare them to in the DFT calculation, instead of matching every band, which is almost never successfully achievable for moderately correlated multi-orbital systems. In response to the Reviewer's comment regarding "excellent agreement," we agree that it is better to be modified to "reasonable overall agreement."

Nevertheless, we respect the Reviewer's opinion and answer each of the Reviewer's questions regarding the detailed band structure, starting with the renormalized bands at $\bar{\Gamma}$. We would like to point out that in the figure attached by the Reviewer, right panel (d) (where the Reviewer referred to bands #4, #5 and #6) was not the revised version from the last round, but from the earlier version of our manuscript. We have made substantial revisions to Fig. 4 in our last revision and hope that the Reviewer can appreciate our efforts. As we have discussed in the previous response, we performed additional SPR-KKR calculations, a different type of DFT calculations complementary to VASP. These new results are incorporated in the new Fig. 4, attached below for the Reviewer's convenience. They facilitated a significantly improved comparison between ARPES data and DFT calculations, as we discuss below and also in the revised manuscript.

Figure R5 (revised Fig. 4) Strong band renormalization in $d_{xy} + d_{x^2-y^2}$ spin majority channel (full caption in revised manuscript Fig. 4)

In Fig. R5 a-c, a direct comparison of the $\bar{K} - \bar{\Gamma} - \bar{K}$ cut between ARPES, SPR-KKR and VASP ($k_z = \pi$) is presented. Both calculations show three electron-like bands (two of them marked by yellow, red boxes) and one hole-like band (green box) close to $\bar{\Gamma}$, where, noticeably, the electron-like bands are renormalized by a factor of 1.7 in SPR-KKR compared with VASP, and the hole-like band is shifted towards E_F but not

much renormalized. We note that in the ARPES cut, there are also two electron-like bands (yellow, red boxes) with similar k_F but further strongly renormalized from both calculations. The third electron-like band is possibly above E_F (band #6 as the Reviewer referred to). The hole-like band (band #4 as the Reviewer referred to) in ARPES data (green box) corresponds to the hole-like band in both calculations, which is again shifted towards E_F but not renormalized. Furthermore, the bands marked by the dark and light blue boxes match the ones in both calculations, which are part of the Dirac bands.

Based on all the above evidence and hints, it is conceivable that the two electron-like bands are selectively renormalized, while the hole-like band is shifted towards E_F but not renormalized. Remarkably, the spin-and-orbital channels that each of these bands belong to (see Fig. 3) suggest a self-consistent spin-and-orbital selective band renormalization scenario. Specifically, the electron-like bands ($d_{xy} + d_{x^2-y^2}$ spin majority) are strongly renormalized, whereas the hole-like band ($d_{xz} + d_{yz}$ spin minority) and the Dirac bands ($d_{xy} + d_{x^2-y^2}$ spin minority) are much less renormalized.

As the Reviewer has mentioned, we shall justify the assignment of the renormalized electron-like bands at $\bar{\Gamma}$ to the bands at $k_z = \pi$. Here we provide photon energy ($h\nu$) dependence measurement of the FeSn film along the $\bar{\Gamma} - \bar{K} - \bar{M}$ cut (Fig. R6, now Fig. S9). We utilize the feature at $\bar{\Gamma}$ at -1.15 eV (yellow dashed line in Fig. R6 a) that shows a clear k_z -dependence to map out the $h\nu$ - k_z relationship. As shown in Fig. R6 e, the helium-lamp photon energy (21.2 eV) corresponds to the red arc, which is closer to $k_z = \pi$ than $k_z = 0$. In particular, the 2nd BZ center Γ ($k_x = 1.37 \text{ \AA}^{-1}$) on the arc, where we observe the renormalized electron-like bands, is extremely close to $k_z = \pi$. Therefore, it is reasonable to attribute the majority of the bands measured with helium-lamp near the 2nd BZ center to $k_z = \pi$, while bands away from $k_z = \pi$ can also have some contributions in the measurements in the 1st BZ.

We have added a discussion on photon energy dependence measurement in Supplementary Note 3, which is referenced in the main text.

Figure R6 (now Fig. S9) Photon energy dependence measurement on FeSn film.

We would also refer the Reviewer to Fig. R5 g-i, where a comparison of the constant energy contours (CECs) is displayed. A remarkable similarity between the CECs from ARPES data and those from the two DFT calculations at $k_z = \pi$ is evident after the renormalization of energy scale is considered, providing strong evidence that not only along the high-symmetry directions, but in the whole momentum space, the ARPES data show remarkable resemblance with SPR-KKR and VASP calculations.

Next, we discuss the bands near \bar{K} and \bar{M} , i.e. the bands #1, #2 and #3 as the Reviewer referred to. We can now have a better comparison of ARPES data and DFT with the knowledge of k_z -dependence. We note that the Dirac cones at \bar{K} is mostly of the $d_{xy} + d_{x^2-y^2}$ character, which has minimal k_z -dependence. Indeed, the ARPES Dirac bands match the DFT ones well. However, the bands related to VHS (band #3 and band #1 defined by the Reviewer) are of the $d_{xz} + d_{yz}$ character, and shows a considerable k_z -dependence (Fig. R7, reproduced from Fig. 3d). While the VHS bands are well preserved along Γ -M, they are hybridized and no longer show VHS behavior along A-L. Hence the VHS nature is highly k_z -dependent. As shown in Fig. R6 e, the $h\nu = 21.2$ eV arc places the 1st BZ boundary (\bar{K} and \bar{M}) between $k_z = 0$ and π . As a result, band #1 may correspond to the feature at -0.4 eV shifted going away from $k_z = 0$, or possibly other bands. The VHS (band #3) is also restricted to $k_z = 0$ (Fig. R7), therefore not showing up in our ARPES data away from $k_z = 0$. Nonetheless, we emphasize that the orbital-selective renormalization is demonstrated by the contrast between the Dirac bands and the electron bands at the $\bar{\Gamma}$ point. A complete one-to-one match of every single band between first principles calculations and measurement depends on the incorporation of

correlation effects into the theoretical calculations. We hope that our work will motivate future theory work in this direction to understand the missing ingredients.

Figure R7 (reproduced from Fig. 3d) $d_{xz} + d_{yz}$ spin minority projected bands.

I acknowledge again the value and interest of the work, but to me the conclusions are not supported at this stage and I advise against publication.

Reply: We sincerely thank the Reviewer for taking the time to review our manuscript and acknowledging the value and interest of our work. We believe that in the revised manuscript the conclusions are supported and the manuscript meets the criteria for publication. Importantly, we hope that the Reviewer agrees with the value our work brings to the understanding of kagome family in the discussion of correlation effects and the nature of the magnetism.

Reviewer #3 (Remarks to the Author):

The study of flat band-induced metallic magnetism has been a vibrant area of research ever since Mielke's pioneering theoretical prediction over 30 years ago of a ferromagnetic ground state in the Hubbard model on the Kagome lattice. This field has gained renewed attention with the recent discoveries of superconductivity and orbital magnetism in twisted bilayer graphene, as well as unconventional charge density waves and superconductivity in metallic bulk Kagome systems. Correlated flat bands are now recognized as one of the most exciting and timely topics in modern condensed matter physics.

In this context, the present work offers a significant contribution by providing the first detailed study of the temperature-dependent evolution of the electronic structure in the well-studied Kagome flat band magnet FeSn. The authors present compelling evidence that the magnetism in this system is primarily due

to local moments and Heisenberg ferromagnetism, rather than itinerant electrons leading to Stoner-type ferromagnetism. This finding is likely to play a crucial role in shaping future theoretical models that aim to accurately describe metallic Kagome magnets, such as development of compact molecular orbitals that respect the topological constraints of the flat bands. It is clear that this research will be of great interest to a wide audience, and I strongly recommend its publication in Nature Communications.

Reply: We sincerely thank the Reviewer for recognizing the wide interest in the magnetic kagome flat band systems and the significance of our work. We very much appreciate the Reviewer's strong recommendation of our manuscript for publication.

That said, while I find the interpretation of the temperature evolution quite convincing, I do have some remaining concerns regarding the second major finding of the paper, specifically related to the interpretation of the flower-like bands near the $G_{\bar{}}$ points. This was already mentioned by the other referees, and I am not yet fully convinced by the response that the authors have provided.

Reply: We thank the Reviewer for finding that our interpretation of the temperature evolution to be quite convincing and for confirming our association of this behavior with local moment physics.

The new analysis presented in Figures 4g-i provides strong arguments for interpreting the flower-like bands as renormalized bulk bands. However, a potential inconsistency arises when comparing the flower-like Fermi surface in Fig. 4g, which is attributed to the Sn termination, with the "Kagome" termination shown in Fig. S11. In the latter case, the six petals of the Fermi surface are absent. This behavior aligns with the termination-dependent flower-like Fermi surface reported by Kang et al. (Nat. Mater. 19, 163 (2020)), where the six petals appear only for the Sn-terminated surface (Fig. 2c). If these petals are indeed bulk states, their absence on the "Kagome termination" is puzzling. I suggest that the authors address this apparent inconsistency in the discussion section. Additionally, mentioning that photon energy-dependent measurements could be conducted in future studies to further explore the dimensionality (and thus potential surface character) of the flower-like Fermi surface would be valuable.

Reply: We thank the Reviewer for raising this important point regarding the flower-like bands and the absence of the flower petals on the kagome termination in Kang et al. Nat. Mater. 19, 163 (2020). This is indeed a puzzle for which we can only think of two possibilities: i) this is a surface state, and ii) there is an intrinsic difference between thin films and single crystals regarding this feature. We have done everything that we can to address the first possibility.

First, we would like to reiterate that the extreme robustness of the flower-like bands strongly suggests against them being a surface state. The extreme robustness includes surviving thermal cycling up to 385 K, and also surviving the Se-capping and de-capping process. In contrast, typical surface states do not stand thermal cycling. An example is provided in Fig. R8, reproduced from Moore et al., PRB 106, 115141 (2022).

In addition, we have also carried out slab calculations that do not show the surface states to exhibit any strong renormalization compared to bulk bands. Hence surface state cannot be the reason for these extremely renormalized bands. All the above evidence in our data leads to the conclusion that the flower-like strongly renormalized bands are bulk states. Therefore this leaves us with the possibility that there is a subtle difference in these bands between thin film and bulk crystals. To this end, we have examined the constant energy contours associated with these bands from DFT. We note that whether these bands exhibit circular pocket or flower petals depends sensitively on how the multiple bands hybridize near Γ . For example, the constant energy contours at $k_z=0$ are circular while they are flower-like at π . Hence it is possible that the details of the existence of the petals depends sensitively on the surface potential, dimensionality and photon energy.

With that said, we do note that there are two observable electron-like bands at the $\bar{\Gamma}$ point that exhibit such strong renormalization. Specifically, inside the flower-like pocket there is another circular electron pocket at $\bar{\Gamma}$ observed on the FeSn thin film (Fig. 4g left panel), which is strongly renormalized by a similar factor (Fig. 4j). This inner pocket is also observed on the kagome termination in the single crystal data, confirming its bulk nature (Fig. S14). In total, all of the electron bands at the $\bar{\Gamma}$ point observed in thin film and single crystals exhibit this strong renormalization, and hence we conclude that the selective renormalization is generic to these features in both single crystals and thin films. We appreciate very much the Reviewer's constructive comment regarding this point and have now explicitly added a discussion of this aspect to the main text:

“Here we note that the prominent flower shape of the electron pocket observed on thin films does not seem to be observed on FeSn single crystals. Although our data provide strong evidence that the remarkably renormalized bands are bulk states, we note the apparent conflicting evidence for them being surface states in a previous study on FeSn single crystals⁴³. To resolve this, future work including photon energy dependent measurements as well as studies to explore the effect of dimensionality would be desired. Nevertheless, the robustness of the strong renormalization of these states are observed on all electron bands measured on thin films as well as single crystals.”

Furthermore, the strong renormalization observed in the bands near the Fermi level raises some questions.

Is it really plausible to have such significant renormalization while still maintaining sharp bands near the Fermi level? A large renormalization typically implies a large real part of the self-energy, which, through the Kramers-Kronig relations, should correspond to a substantial imaginary part of the self-energy. However, the lifetime broadening here appears minimal. The authors might consider discussing the relationship between strong renormalization and lifetime broadening in other materials to clarify this point.

Reply: We thank the Reviewer for this important comment. Actually this kind of strong orbital-selective behavior has been observed in both the ruthenates and the iron-based superconductors. In the iron-chalcogenide family of Fe(Te,Se), the d_{xy} orbital could exhibit mass enhancement up to 40 (Commun. Phys. 5, 29 (2022)), while in the ruthenates this could be up to 25 (New J. Phys. 15 063029 (2013)), while still maintaining sharp quasiparticles. Theoretically one way to understand this phenomenology is the concept of Hund's metal, applicable to systems away from the Mott insulating state at half filling. In such multi-orbital systems, different orbitals exhibit Fermi liquid behavior in a fully coherent state at low temperatures, then undergo a coherent-incoherent crossover above a characteristic coherence temperature scale T^* where their lifetime is broadened to an extent that they no longer qualify as well-defined quasiparticles. This temperature scale is different for different orbitals due to the distinct strength of electron correlations between these orbitals. In FeSn, as the near- E_F states are also Fe 3d, it is highly likely that similar physics is at play and that the temperature scale that we observe where the strongly renormalized electron bands lose coherence is due to similar physics. We very much appreciate the Reviewer's suggestion to add a discussion of this and especially include discussion of such behavior in other material systems:

"The selective renormalization together with the rapid depletion of the spectral weight of the renormalized bands with increasing temperature is reminiscent of the orbital-selective correlations observed in multi-orbital systems, most prominently reported in the ruthenates⁵³⁻⁵⁵ and the iron-based superconductors⁵⁶⁻⁶². In these systems deemed Hund's metals, bands associated with a particular orbital is strongly renormalized already at low temperatures, with mass enhancement ranging from 25 in the case of ruthenates and up to 40 in the iron chalcogenides, but retaining relatively well-defined electronic states. Above a characteristic temperature scale, these strongly renormalized orbitals lose coherence and are no longer well-defined quasiparticles. Theoretically, this behavior has been understood to arise from a combination of Hund's coupling J and Coulomb interaction U . For such multi-orbital systems away from half-filling, the occupation of different orbitals could vary, with some closer to half-filling. As demonstrated by both slave-boson calculations and dynamical mean field theory calculations, these orbitals are typically more strongly renormalized and exhibit a lower coherence temperature scale than other orbitals, where photoemission measurements would observe a spectral weight depletion for these orbitals as a function of temperature at a much lower temperature than other less renormalized orbitals⁶³⁻⁶⁸. Here in the case of FeSn, both characteristic strong selective renormalization as well as coherence depletion are observed for the electron bands near $\bar{\Gamma}$."

Beyond this primary concern, I have a few minor comments that the authors might find helpful:

1. The authors mention that their films are primarily Sn-terminated. It would be beneficial to provide the experimental evidence that supports this conclusion.

Reply: We made this conclusion based on the spatial mapping of the Sn core levels on the thin film sample. As shown in Fig. R10 (now Fig. S13), the integrated intensity of the cuts enclosing the Sn 4d core levels is homogeneous over the scanned region (Fig. R10 a). The core-level spectra show the Sn surface peaks uniformly (Fig. R10 b). We have now added this to the Supplementary Materials.

Figure R10 (also Fig. S13) Sn 4d core level spatial mapping on the FeSn film sample.

2. Have the authors studied the thicker films used for magnetization measurements with ARPES? If not, do they anticipate any differences in the electronic structure between thin and thick films?

Reply: The nominal thickness of the film used for magnetization measurements was 300 nm, which facilitated the measurement of the weak signal from the antiferromagnetic order of FeSn. We did not measure the same film with ARPES. Nonetheless, we have measured a film with a nominal thickness of 63 nm, doubling the film thickness shown in the manuscript. We did not find noticeable difference in the band structure of the 63 nm and 30 nm films, as shown in Fig. R11. Since these films are more than 60 unit cells thick ($c = 0.446$ nm), we consider them as in the bulk regime and anticipate the same electronic structure as a bulk crystal.

Figure R11 Fermi surface map and the K-M-K cut of a 63 nm thick FeSn film.

3. In Fig. 2a, the peak in the DOS near E_F in the PM state is not symmetrically split around the Fermi level in the AFM phase. Is there a straightforward explanation for this?

Reply: The ferromagnetic phase indicates an imbalance in the number of spin-up and spin-down states, but there is no obvious reason or constraint that the exchange splitting should be symmetric between the opposite spin species because the electronic states and therefore the DOS is not symmetric about the kagome flat bands. The magnitude of the splitting might depend on the material. In this case, it is possible that a symmetric splitting would create too large of an imbalance between the number of spin-up and spin-down states due to the large density of states of the flat bands, which is not allowed physically.

4. The sentence, “Although most bands in the DFT calculations have a good match in the ARPES data (Fig. 2d,j), the calculated spin majority electron pocket along $K_{\text{bar}}-G_{\text{bar}}-K_{\text{bar}}$ within 0.5 eV below E_F seems to have no experimental counterpart (Fig. 2d),” is somewhat unclear. I don’t see an electron pocket in Fig. 2d.

Reply: We apologize for the confusion. We replaced “electron pocket” with “electron-like band” in the revised sentence.

5. Do the authors have any insights into why the bands corresponding to the vHS below E_F are not observed?

Reply: This particular VHS is of the $d_{xz} + d_{yz}$ spin minority character, as shown by the orbital-spin projected DFT calculations, and exhibits strong k_z -dependence. Specifically, at A-L, they no longer exhibit the VHS characteristic saddle point behavior. At the photon energy of the helium lamp, the k_z is close to $k_z=\pi$ at the center of the second BZ (Supplementary Note 3). Hence it is likely that the VHS band structure is already not preserved here. We have added a discussion of this to the main text:

“Below E_F , another pair is located at -0.2 eV and -0.4 eV at the M point, dominantly of the $d_{xz} + d_{yz}$ orbital. It is important to note that as d_{xz} and d_{yz} are three-dimensional orbitals and these VHS exhibit strong k_z -dispersion such that along A-L they no longer preserve their saddle-point behavior. In the measured dispersions, a band is indeed observed near the location of the higher VHS, but it is hole-like along both $\bar{\Gamma} - \bar{M}$ and $\bar{M} - \bar{K}$, and is therefore not a VHS, likely due to this strong k_z -dispersion.”

6. In Fig. 3a, the green boxes appear quite large. Perhaps arrows pointing to the narrow bands at Gamma would be more precise.

Reply: We follow the Reviewer’s suggestion to change the green boxes to green arrows pointing to the narrow bands at Γ .

7. Is it known why strongly renormalized bands might exhibit a more pronounced temperature dependence in their coherence factor compared to more weakly renormalized bands? If so, a brief comment on this would be appreciated.

Reply: The situation here is reminiscent of the temperature-induced orbital-selective Mott crossover observed in ruthenates and the Fe-based superconductors, for example, as shown in Fig. R12 reproduced from Yi et al., PRL 110, 067003 (2013). In that case, the d_{xy} orbital is selectively strongly-renormalized in the correlated metallic phase with a small Z coherence factor and experiences a crossover into a localized phase (loses its coherence) with increasing temperature. Two theoretical approaches have explained this behavior. In the slave-spin calculation incorporating Coulomb interactions U and Hund's J, while the coherence factors for all orbitals decrease at this transition, the one for d_{xy} becomes zero due to its smaller initial coherence factor. Alternatively, dynamical mean field theory calculations have shown that in metals away from half-filling (known as Hund's metals), a coherence temperature scale exists below which the renormalized electronic states of certain orbitals exhibit Fermi liquid behavior, but lose their coherence above this temperature scale. This temperature scale is lower for more strongly renormalized orbitals (PHYSICAL REVIEW B 86, 195141 (2012); PRL 106, 096401 (2011)). Here in FeSn, the $d_{xy} + d_{x^2-y^2}$ spin majority bands are also likely to be closer to an orbital (and spin)-selective Mott transition, and therefore a more rapid loss of their coherence upon warming. We thank the Reviewer for this suggestion and have added a comment in the manuscript to address this:

“As demonstrated by both slave-boson calculations and dynamical mean field theory calculations, these orbitals are typically more strongly renormalized and exhibit a lower coherence temperature scale than other orbitals, where photoemission measurements would observe a spectral weight depletion for these orbitals as a function of temperature at a much lower temperature than other less renormalized orbitals^{63–68}. Here in the case of FeSn, both characteristic strong selective renormalization as well as coherence depletion are observed for the electron bands near $\bar{\Gamma}$.”

I hope that these comments are helpful and constructive, as my intention is to further improve the already impressive quality of the paper. I look forward to seeing the final version of this exciting work.

Reply: We sincerely thank the Reviewer again for praising the quality of our manuscript and for all of the really helpful and constructive suggestions and discussions to help us improve our manuscript.

We conclude the response letter with a list of changes made to the manuscript:

Main Text:

1. Data showing temperature dependence of the Dirac cones up to 386 K $> T_N$, and the analysis on MDC fitting to demonstrate the small downward shift upon warming.
2. Photon-energy dependence data for $\hbar\nu$ - k_z relationship, and justification of the comparison of the strongly renormalized electron-like bands with DFT bands at $k_z=\pi$.
3. Addressing the missing VHS possibly due to k_z being away from $k_z=0$.
4. Addressing the conflicting evidence for the flower-like pocket being surface states, and mentioning that future work would be desired regarding this.
5. Discussion on other materials exhibiting strong selective band renormalization, depletion of quasiparticle coherence with increasing temperature, and different theoretical models on these systems.
6. Evidence for the Sn termination of the film samples.
7. Minor modifications to the figures such as arrows and boxes.

Supplementary Materials:

Added new Fig. S7 to show temperature dependence of the Dirac cone and associated MDC analysis.

Added new Fig. S9 and Supplementary Note 3 on the photon energy dependence.

Added new Fig. S13 on the spatial mapping of the core levels measured on FeSn thin film.

REVIEWERS' COMMENTS

Reviewer #1 (Remarks to the Author):

In the present resubmission, the authors have extended the temperature range where the Dirac cone is measured (Fig. 2k) and added a characterization of k_z dependence (Fig. S9), which I consider as significant improvements.

- Persistent magnetic splitting : Even if the Dirac cone feature is weak and broad at 386K, I find it sufficient to believe the Dirac cone is not moving drastically. Personally, I still find the temperature range above T_N (a mere 15K) too small to safely conclude on a persistent magnetic splitting. However, I acknowledge, as I did before, that the measurement is not easy.

- Orbital selective renormalization : I am not asking for a perfect match between the measurement and the calculation, but a sufficient agreement to be reasonably sure of the assignment of the « flower shape » FS to the spin majority $d_{xy}-d_{x^2-y^2}$, hence of the conclusion about orbital selective renormalization. To me, this is not the case here. To avoid a fuzzy discussion, I have given in my previous report some precise points of disagreement between calculation and experiment, but the authors reply to them very selectively (the case of Fig.3 is almost ignored). I appreciate however the photon energy measurement reporting in Fig. S9, pointing to a k_z value near $k_z=1$ in BZ2 (I suppose there is a mistake in the k_z-k_x map indicating K instead of M). I also find the discussion of surface state problem better done than in first versions. The calculation part itself is rather intriguing, but it is frustrating that there is so little discussion of the effect. It is only written at the end of page 8 « Interestingly, spin also participates in the selectivity as another degree of freedom ». The entire paragraph before about Fe-pnictides does not bring anything useful in my opinion.

I still have the impression that more data (especially those the authors probably have as a function of photon energy) would be needed to safely conclude about the 2 points raised by this paper. Although I do not support publication in Nature communications in the present stage, I would not oppose it, as the manuscript is more convincing now and it could be argued that, in this active and competitive field, the ideas the authors develop can be stimulating for many others.

Reviewer #3 (Remarks to the Author):

The authors have addressed all of my comments and remarks to my satisfaction. I have also reviewed the comments from Reviewer #1 and the author's replies, and believe that the authors have addressed these comments as well. I therefore recommend publication.

Reviewer #1 (Remarks to the Author):

In the present resubmission, the authors have extended the temperature range where the Dirac cone is measured (Fig. 2k) and added a characterization of k_z dependence (Fig. S9), which I consider as significant improvements.

Reply: We thank the Reviewer for acknowledging our additional data as significant improvements.

- Persistent magnetic splitting: Even if the Dirac cone feature is weak and broad at 386K, I find it sufficient to believe the Dirac cone is not moving drastically. Personally, I still find the temperature range above T_N (a mere 15K) too small to safely conclude on a persistent magnetic splitting. However, I acknowledge, as I did before, that the measurement is not easy.

Reply: We are delighted that the Reviewer finds the temperature dependence of the Dirac cone sufficient to support our conclusion. We are also thankful to the Reviewer for acknowledging the difficulty of our measurements.

- Orbital selective renormalization: I am not asking for a perfect match between the measurement and the calculation, but a sufficient agreement to be reasonably sure of the assignment of the « flower shape » FS to the spin majority $d_{xy}-d_{x^2-y^2}$, hence of the conclusion about orbital selective renormalization. To me, this is not the case here. To avoid a fuzzy discussion, I have given in my previous report some precise points of disagreement between calculation and experiment, but the authors reply to them very selectively (the case of Fig.3 is almost ignored). I appreciate however the photon energy measurement reporting in Fig. S9, pointing to a k_z value near $k_z=1$ in BZ2 (I suppose there is a mistake in the k_z-k_x map indicating K instead of M). I also find the discussion of surface state problem better done than in first versions. The calculation part itself is rather intriguing, but it is frustrating that there is so little discussion of the effect. It is only written at the end of page 8 « Interestingly, spin also participates in the selectivity as another degree of freedom ». The entire paragraph before about Fe-pnictides does not bring anything useful in my opinion.

Reply: We thank the Reviewer for acknowledging the photon energy dependence data for assigning k_z . We note that there is no mistake in the k_z-k_x map in Supplementary Figure 9e, as the photon energy measurement was taken along the $\bar{\Gamma} - \bar{K} - \bar{M}$ direction. Since the 2nd BZ $\bar{\Gamma}$ is located along the $\bar{\Gamma} - \bar{M}$ direction instead of $\bar{\Gamma} - \bar{K} - \bar{M}$, we chose to plot the in-plane BZs the way shown in Supplementary Figure 9e. We are also thankful to the Reviewer for finding our surface state discussion better done, and for finding the calculation part intriguing. To fully understand the selective renormalization effect in the calculations is itself a challenging theoretical problem, which is beyond the scope of this primarily experimental work. We believe that our comprehensive ARPES data and DFT calculations with different approaches provide reasonable grounds for the assignment of the bands. In particular, we have addressed the Reviewer's questions regarding the match of the bands in Fig. 3 to our best, mentioning that the k_z -dependence of the $d_{xz} + d_{yz}$ bands possibly cause the mismatch to the ARPES data taken at k_z away from 0 or π .

I still have the impression that more data (especially those the authors probably have as a function of photon energy) would be needed to safely conclude about the 2 points raised by this paper. Although I do not support publication in Nature communications in the present stage, I would not oppose it, as the manuscript is more convincing now and it could be argued that, in this active and competitive field, the ideas the authors develop can be stimulating for many others.

Reply: We thank the Reviewer for not opposing the publication of our manuscript, and for finding our results more convincing and stimulating.

Reviewer #3 (Remarks to the Author):

The authors have addressed all of my comments and remarks to my satisfaction. I have also reviewed the comments from Reviewer #1 and the author's replies, and believe that the authors have addressed these comments as well. I therefore recommend publication.

Reply: We sincerely thank the Reviewer for recommending publication.